# Evolution of irreversible somatic differentiation

**Yuanxiao Gao[1], Hye Jin Park[1,2,3], Arne Traulsen[1], Yuriy Pichugin[1†]\***

[1]Max Planck Institute for Evolutionary Biology, Plön, Germany; [2]Asia Pacific Center for Theoretical Physics, Pohang, Republic of Korea; [3]Department of Physics, POSTECH, Pohang, Republic of Korea

**Abstract** A key innovation emerging in complex animals is irreversible somatic differentiation: daughters of a vegetative cell perform a vegetative function as well, thus, forming a somatic lineage that can no longer be directly involved in reproduction. Primitive species use a different strategy: vegetative and reproductive tasks are separated in time rather than in space. Starting from such a strategy, how is it possible to evolve life forms which use some of their cells exclusively for vegetative functions? Here, we develop an evolutionary model of development of a simple multicellular organism and find that three components are necessary for the evolution of irreversible somatic differentiation: (i) costly cell differentiation, (ii) vegetative cells that significantly improve the organism's performance even if present in small numbers, and (iii) large enough organism size. Our findings demonstrate how an egalitarian development typical for loose cell colonies can evolve into germ-soma differentiation dominating metazoans.

**\*For correspondence:**
pichugin@evolbio.mpg.de

**Present address:** [†]Department of Ecology and Evolutionary Biology, Princeton University, Princeton, United States

**Competing interests:** The authors declare that no competing interests exist.

## Introduction

In complex multicellular organisms, different cells specialise to execute different functions. These functions can be generally classified into two kinds: reproductive and vegetative. Cells performing reproductive functions contribute to the next generation of organisms, while cells performing vegetative function contribute to sustaining the organism itself. In unicellular species and simple multicellular colonies, these two kinds of functions are performed at different times by the same cells – specialization is temporal. In more complex multicellular organisms, specialization transforms from temporal to spatial (*Mikhailov et al., 2009*), where groups of cells focused on different tasks emerge in the course of organism development.

Typically, cell functions are changed via differentiation, such that a daughter cell performs a different function than the maternal cell. The vast majority of metazoans feature a very specific and extreme pattern of cell differentiation: any cell performing vegetative functions forms a somatic lineage, that is, producing cells performing the same vegetative function – somatic differentiation is irreversible. Since such somatic cells cannot give rise to reproductive cells, somatic cells do not have a chance to pass their offspring to the next generation of organisms. Such a mode of organism development opened a way for deeper specialization of somatic cells and consequently to the astonishing complexity of multicellular animals. Outside of the metazoans – in a group of green algae *Volvocales* serving as a model species for evolution of multicellularity – the emergence of irreversibly differentiated somatic cells is the hallmark innovation marking the transition from colonial life forms to multicellular species (*Kirk, 2005*).

While the production of individual cells specialized in vegetative functions comes with a number of benefits (*Grosberg and Strathmann, 2007*), the development of a dedicated vegetative cell lineage that is lost for organism reproduction is not obviously a beneficial adaptation. From the perspective of a cell in an organism, the guaranteed termination of its lineage seems the worst possible evolutionary outcome for itself. From the perspective of an entire organism, the death of somatic

cells at the end of the life cycle is a waste of resources, as these cells could in principle become parts of the next generation of organisms. For example, exceptions from irreversible somatic differentiation are widespread in plants (*Lanfear, 2018*) and are even known in simpler metazoans among cnidarians (*DuBuc et al., 2020*) for which differentiation from vegetative to reproductive functions has been reported. Therefore, the irreversibility of somatic differentiation cannot be taken for granted in the course of the evolution of complex multicellularity.

Terminal differentiation is a type of cell differentiation different from irreversible cell differentiation. Unlike irreversibly differentiated cells who are capable of cell division, terminally differentiated cells lose the ability to divide. Terminally differentiated cells often perform tasks too demanding to be compatible with cell division. For example heterocysts of cyanobacteria perform nitrogen fixation, which requires anaerobic conditions, therefore these cells are very limited in resources and do not divide. In the scope of this study, we do not consider terminal differentiation but focus on somatic cells that are able to divide while being part of an organism (or cell colony) but not able to grow into a new organism, that is, irreversible somatic differentiation.

The majority of the theoretical models addressing the evolution of somatic cells focuses on the evolution of cell specialization, abstracting from the developmental process how germ (reproductive specialists) and soma are produced in the course of the organism growth. For example, a large amount of work focuses on the optimal distribution of reproductive and vegetative functions in the adult organism (*Michod, 2007*; *Willensdorfer, 2009*; *Rossetti et al., 2010*; *Rueffler et al., 2012*; *Ispolatov et al., 2012*; *Goldsby et al., 2012*; *Solari et al., 2013*; *Goldsby et al., 2014*; *Amado et al., 2018*; *Tverskoi et al., 2018*). However, these models do not consider the process of organism development. Other work takes the development of an organism into account to some extent: In *Gavrilets, 2010*, the organism development is considered, but the fraction of cells capable of becoming somatic is fixed and does not evolve. In *Erten and Kokko, 2020*, the strategy of germ-to-soma differentiation is an evolvable trait, but the irreversibility of somatic differentiation is taken for granted. In *Rodrigues et al., 2012*, irreversible differentiation was found, but both considered cell types pass to the next generation of organisms, such that the irreversible specialists are not truly somatic cells in the sense of evolutionary dead ends. Finally, in *Cooper and West, 2018* a broad scope of cell differentiation patterns has been investigated in the context of evolution of cooperation. However, irreversible somatic differentiation was not considered in the study. Hence, the theoretical understanding of the evolution of irreversibly differentiated somatic cell lines is limited so far.

In the present work, we developed a theoretical model to investigate conditions for the evolution of the irreversible somatic differentiation. In the model, we suppose there are two cell types: germ-role and soma-role, where only germ-role cells pass to the next generation of organisms while soma-role cells are responsible for vegetative functions. Both germ-role cells and soma-role cells can divide and they may switch to each other during growth. In our model, we incorporate factors including (i) costs of cell differentiation, (ii) benefits provided by presence of soma-role cells, (iii) maturity size of the organism. We ask under which circumstances irreversible somatic differentiation is a strategy that can maximize the population growth rate compared to strategies in which differentiation does not occur or somatic differentiation is reversible.

## Model

We consider a large population of clonally developing organisms composed of two types of cells: germ-role and soma-role. The roles differ in the ability to survive beyond the end of the organism life cycle: soma-role cells die at the end, while germ-role cells continue to live. Each organism is initiated as a single germ-role cell. In the course of the organism growth, germ-role cells may differentiate to give rise to soma-role cells and vice versa, see *Figure 1A,B*. After $n$ rounds of synchronous cell divisions, the organism reaches its maturity size of $2^n$ cells. Immediately upon reaching maturity, the organism reproduces: germ-role cells disperse and each becomes a newborn organism, while all soma-role cells die and are thus lost, see *Figure 1A*. We assume that soma-role cells are capable to accelerate growth: an organism containing more somatic cells grows faster, so having soma-role cells during the life cycle is beneficial for the organism.

To investigate the evolution of irreversible somatic differentiation, we consider organisms in which the functional role of the cell (germ-role or soma-role) is not necessarily inherited. When a cell divides, the two daughter cells can change their role, leading to three possible combinations: two

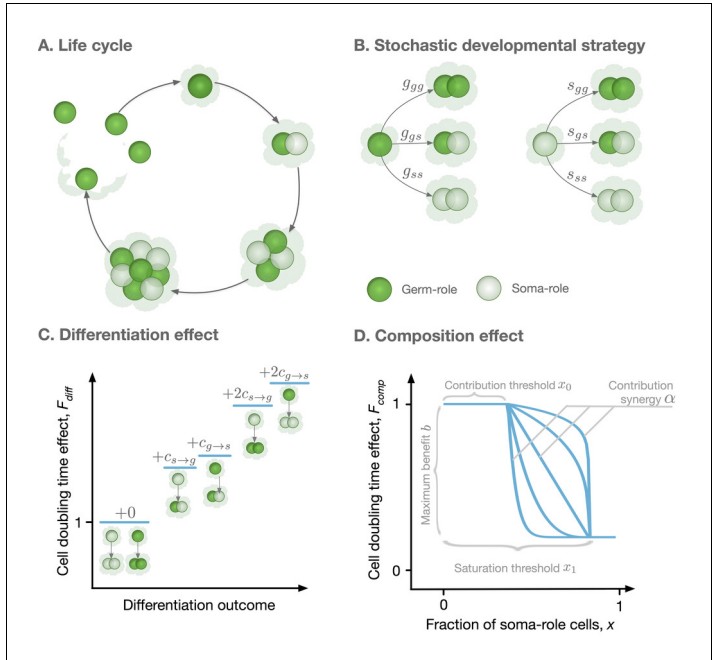

**Figure 1.** Model overview. (A) The life cycle of an organism starts with a single germ-role cell. In each round, all cells divide and daughter cells can differentiate into a role different from the maternal cell's role. When the organism reaches maturity, it reproduces: each germ-role cell becomes a newborn organism and each soma-role cell dies. (B) Change of cell roles is controlled by a stochastic developmental strategy defined by probabilities of each possible outcomes of a cell division. (C) Differentiation of cells requires an investment of resources and, thus, slows down the organism growth. Each cell differentiation event incurs a cost ($c_{s \to g}$ or $c_{g \to s}$). (D) The growth contribution of somatic cells is controlled by a function that decreases the doubling time with the fraction of somatic cells. The form of this function is controlled by four parameters, $x_0$, $x_1$, $\alpha$, and $b$.

germ-role cells, one germ-role cell plus one soma-role cell, or two soma-role cells. We allow all these outcomes to occur with different probabilities, which also depend on the parental type, see *Figure 1B*. If the parental cell had the germ-role, the probabilities of each outcome are denoted by $g_{gg}$, $g_{gs}$, and $g_{ss}$ respectively. If the parental cell had the soma-role, these probabilities are $s_{gg}$, $s_{gs}$, and $s_{ss}$. Altogether, six probabilities define a stochastic developmental strategy $D = \left(g_{gg}, g_{gs}, g_{ss}; s_{gg}, s_{gs}, s_{ss}\right)$. In our model, it is the stochastic developmental strategy that is inherited by offspring cells rather than the functional role of the parental cell.

To feature irreversible somatic differentiation, the developmental strategy must allow germ-role cells to give rise to soma-role cells ($g_{gg} < 1$) and must forbid soma-role cells to give rise to germ-role cells ($s_{ss} = 1$). All other developmental strategies can be broadly classified into two classes. Reversible somatic differentiation describes strategies where cells of both roles can give rise to each other: $g_{gg} < 1$ and $s_{ss} < 1$. In the strategy with no somatic differentiation, soma-role cells are not produced in the first place: $g_{gg} = 1$, see *Table 1*.

In our model, evolution of the developmental strategy is driven by the growth competition between populations executing different strategies – these populations able to produce more offspring and/or complete their life cycle faster gain a selective advantage. Specifically, we measure

**Table 1.** Classification of developmental strategies.

| Class | $g_{gg}$ | $s_{ss}$ |
|---|---|---|
| Irreversible somatic differentiation | <1 | = 1 |
| Reversible somatic differentiation | <1 | <1 |
| No somatic differentiation | = 1 | irrelevant |

the fitness in the growth competition by the population growth rate in a stationary regime of exponential growth (*Pichugin et al., 2017*; *Gao et al., 2019*). The rate of population growth is determined by the number of offspring produced by an organism (equal to the number of germ-role cells at the end of life cycle) and the time needed for an organism to develop from a single cell to maturity (improved with the number of soma-role cells during the life cycle).

To obtain these growth rates, we simulate the process of the organism growth. Here, we assume that resource distribution among cells is coordinated at the level of the organism: Cells which need more resources will get more, such that cell division is synchronous. In our model, we consider synchronous cell division of organisms and our main results are dependent on this assumption. However, we shortly explore the effects of asynchronous cell division in Appendix G. Any organism is born as a single germ-role cell and passes through $n$ rounds of simultaneous cell divisions. Each round starts with every cell independently choosing the outcome of its division with probability of each outcome given by the developmental strategy ($D$). This step determines what composition will the organism have at the next round of cell division. Then, the length of the cell doubling round ($t$) is computed as a product of two independent effects: the differentiation effect $F_{\mathrm{diff}}$ representing costs of changing cell roles (*Gallon, 1992*) and the organism composition effect $F_{\mathrm{comp}}$ representing benefits from having soma-role cells (*Grosberg and Strathmann, 1998*; *Shelton et al., 2012*; *Matt and Umen, 2016*),

$$t = F_{\mathrm{diff}} \times F_{\mathrm{comp}}. \tag{1}$$

Both $F_{\mathrm{diff}}$ and $F_{\mathrm{comp}}$ are re-calculated at every round of cell division.

The cell differentiation effect $F_{\mathrm{diff}}$ represents the costs of cell differentiation. The differentiation of a cell requires efforts to modify epigenetic marks in the genome, recalibration of regulatory networks, synthesis of additional and utilization of no longer necessary proteins. This requires an investment of resources and therefore an additional time to perform cell division. Hence, any cell, which is about to give rise to a cell of a different role, incurs a differentiation cost $c_{g \to s}$ for germ-to-soma and $c_{s \to g}$ for soma-to-germ transitions (and double of these if both offspring take a role different from the parent), see *Figure 1C*. The differentiation cost is the averaged differentiation cost among all cells in an organism

$$F_{\mathrm{diff}} = 1 + <c> = 1 + \frac{c_{s \to g}(N_{s \to gs} + 2N_{s \to gg}) + c_{g \to s}(N_{g \to gs} + 2N_{g \to ss})}{N}, \tag{2}$$

where $N_{s \to gs}$ is the number of soma-roll cells that produce a germ-role cell and a soma-role cell in a cell division step. $N_{s \to gg}$, $N_{g \to gs}$ and $N_{g \to ss}$ are defined in the analogous way. $N$ is the number of total cells. As organisms undergo synchronous cell division, we have $N = 2^n$ cells after the $n$ th cell division.

The composition effect profile $F_{\mathrm{comp}}(x)$ captures how the cell division time depends on the proportion of soma-role cells $x = s/(s+g)$ present in an organism ($s$ and $g$ are the numbers of soma-role and germ-role cells). In this study, we use a functional form illustrated in *Figure 1D* and given by

$$F_{\mathrm{comp}}(x) = \begin{cases} 1 & \text{for } 0 \leq \mathrm{x} \leq \mathrm{x}_0 \\ 1 - b + b\left(\frac{x_1 - x}{x_1 - x_0}\right) & \text{for } \mathrm{x}_0 < \mathrm{x} < \mathrm{x}_1 \\ 1 - b & \text{for } \mathrm{x}_1 \leq \mathrm{x} \leq 1 \end{cases} \tag{3}$$

With the functional form (3), soma-role cells can benefit to the organism growth, only if their proportion in the organism exceeds the contribution threshold $\mathrm{x}_0$. Interactions between soma-role cells may lead to the synergistic (increase in the number of soma-role cells improves their efficiency), or discounting benefits (increase in the number of soma-role cells reduces their efficiency) to the organism growth, controlled by the contribution synergy parameter $\alpha$. The maximal achievable reduction in the cell division time is given by the maximal benefit $b$, realized beyond the saturation threshold $x_1$ of the soma-role cell proportion. A further increase in the proportion of soma-role cells does not provide any additional benefits. With the right combination of parameters, (3) is able to recover various characters of soma-role cells contribution to the organism growth: linear ($x_0 = 0, x_1 = 1, \alpha = 1$), power-law ($x_0 = 0, x_1 = 1, \alpha \neq 1$), step-functions ($x_0 = x_1$), and a huge range of other scenarios. Previous works have shown that convex (accelerating) performance functions favour cell differentiation

(*Michod, 2006*; *Rueffler et al., 2012*; *Cooper and West, 2018*). The performance functions measure the performance of organisms with respect to different traits, such as fertility and viability. Lately, the form of functions favoring cell differentiation has been extended to be concave (decelerating) by including topological constraints in organisms (*Yanni et al., 2020*). Our model extends the form of performance functions by allowing it has a contribution threshold and saturation threshold.

Once the outcome of all cell divisions is known and the time needed to complete the current cell doubling round is computed, the current round ends and the next starts. The development completes after $n$ rounds. At this stage, the number of germ-role cells (organism offspring number) and the cumulative length of the life cycle are obtained.

In *Gao et al., 2019*, we have shown that the growth rate ($\lambda$) of a population, in which organisms undergo a stochastic development and fragmentation, is given by the solution of

$$\sum_i G_i P_i e^{-\lambda T_i} = 1. \tag{4}$$

Here, $i$ is the developmental trajectory – in our case, the specific combination of all cell division outcomes; $G_i$ is the number of offspring organisms produced at the end of developmental trajectory $i$, equal to the number of germ-role cells at the moment of maturity; $P_i$ is the probability that an organism development will follow the trajectory $i$; $T_i$ is the time necessary to complete the trajectory $i$ – from a single cell to the maturity size of $2^n$ cells.

For a given combination of differentiation costs ($c_{g \to s}$, $c_{s \to g}$) and a composition effect profile (determined by four parameters: $x_0$, $x_1$, $b$, and $\alpha$), we screen through a number of stochastic developmental strategies $D$ and identify the one providing the largest growth rate ($\lambda$) to the population. In this study, we searched for those parameters under which irreversible strategies lead to the fastest growth and are thus evolutionary optimal, see model details in Appendix A.

## Results

### For irreversible somatic differentiation to evolve, cell differentiation must be costly

We found that irreversible somatic differentiation does not evolve when cell differentiation is not associated with any costs ($c_{s \to g} = c_{g \to s} = 0$), see *Figure 2A*. Only reversible differentiation evolves there, see *Figure 2B*. This finding comes from the fact that when somatic differentiation is irreversible, the fraction of germ-role cells can only decrease in the course of life cycle. As a result, irreversible strategies deal with the tradeoff between producing more soma-role cells at the beginning of the life cycle, and having more germ-role cells by the end of it. On the one hand, irreversible strategies which produce a lot of soma-role cells early on, complete the life cycle quickly but preserve only a few germ-role cells by the time of reproduction. On the other hand, irreversible strategies which generate a lot of offspring, can deploy only a few soma-role cells at the beginning of it and thus their developmental time is inevitably longer. By contrast, reversible somatic differentiation strategies do not experience a similar tradeoff, as germ-role cells can be generated from soma-role cells. As a result, reversible strategy allows higher differentiation rates and can develop a high soma-role cell fraction in the course of the organism growth and at the same time have a large number of germ-role cells by the moment of reproduction. Under costless cell differentiation, for any irreversible strategy, we can find a reversible differentiation counterpart, which leads to faster growth: the development proceeds faster, while the expected number of produced offspring is the same, see Appendix 2 for details. As a result, costless cell differentiation cannot lead to irreversible somatic differentiation.

To confirm the reasoning that reversible strategies gain an edge over irreversible strategies by having larger differentiation rates, we asked which reversible and irreversible strategies become optimal at various cell differentiation costs ($c = c_{s \to g} = c_{g \to s}$). At each value of costs, we found evolutionarily optimal developmental strategy for 3000 different randomly sampled composition effect profiles $F_{\text{comp}}(x)$. We found that evolutionarily optimal reversible strategies feature much larger rates of cell differentiation than evolutionarily optimal irreversible strategies, see *Figure 2D*. Even at large costs, where frequent differentiation is heavily penalized, the distinction between differentiation rates of reversible and irreversible strategies remains apparent.

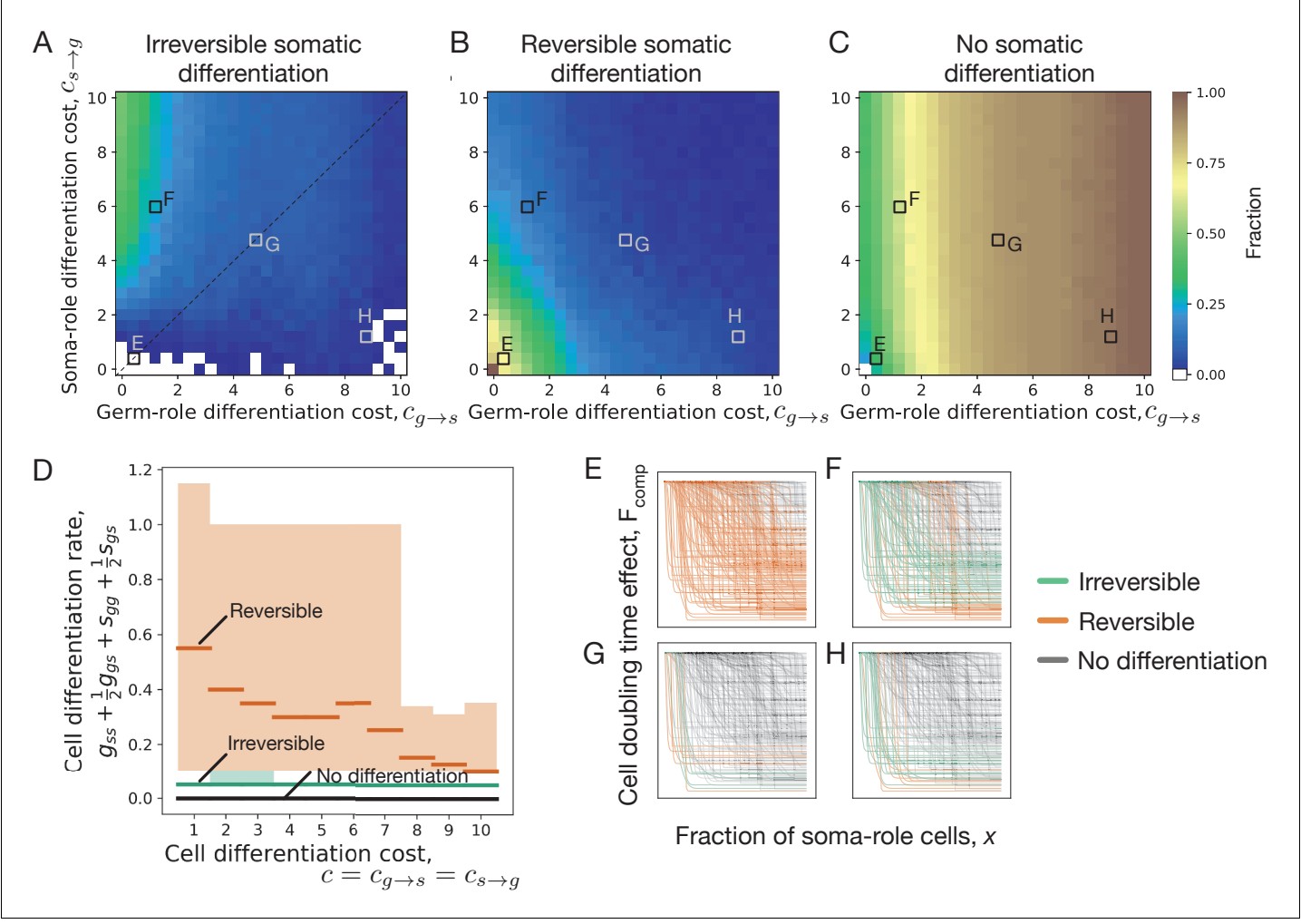

**Figure 2.** Impact of cell differentiation costs on the evolution of development strategies. The fractions of 200 random composition effect profiles promoting irreversible (A), reversible (B), and no differentiation (C) strategies at various cell differentiation costs ($c_{s \to g}$, $c_{g \to s}$). In the absence of costs ($c_{g \to s} = c_{s \to g} = 0$), only reversible strategies were observed. Reversible strategies are prevalent at smaller cell differentiation costs. No differentiation strategies are the most abundant at large costs for germ-role cells ($c_{g \to s}$). Irreversible strategies are the most abundant at large costs for soma-role cells ($c_{s \to g}$). (D) Cumulative cell differentiation rate ($g_{ss} + \frac{1}{2}g_{gs} + s_{gg} + \frac{1}{2}s_{gs}$) in developmental strategies evolutionarily optimal at various differentiation costs ($c_{s \to g} = c_{g \to s}$), separated by class (irreversible somatic differentiation, reversible somatic differentiation, or no somatic differentiation). Thick lines represent median values within each class, shaded areas show 90% confidence intervals. For each cost value, 3000 random profiles are used in this panel. Evolutionary optimal reversible strategies (orange) have much higher rates of cell differentiation than irreversible strategies (green). Consequently, reversible strategies are penalized more under costly differentiation. (E–H) Shapes of composition effect profiles (compare *Figure 1D*) promoting irreversible (green lines), reversible (orange lines), and no differentiation (black lines) strategies at four parameter sets indicated in panel A. The maturity size used in the calculation is $2^{10}$ cells.

We screened through a spectrum of germ-to-soma ($c_{g \to s}$) and soma-to-germ ($c_{s \to g}$) differentiation costs, see *Figure 2A–C*. Irreversible somatic differentiation is most likely to evolve when it is cheap to differentiate from germ-role to soma-role (low $c_{g \to s}$) but it is expensive to differentiate back (high $c_{s \to g}$), see *Figure 2A*. Irreversible strategies are insensitive to high soma-to-germ costs, since soma-role cells never differentiate. At the same time, reversible strategies are heavily punished by high costs of soma-role differentiation.

It is not very surprising to find irreversible differentiation where the differentiation costs are highly asymmetric. However, irreversible strategies are consistently observed in other regions of the costs space, even including these, where the asymmetry is opposite (it is hard to go from germ to soma but easy to return back), see *Figure 2A,H*. To identify what other factors, beyond asymmetric costs,

can lead to evolution of irreversible somatic differentiation, below we focus on the scenario of equal differentiation costs $c_{s \to g} = c_{g \to s} = c$.

## Evolution of irreversible somatic differentiation is promoted when even a small number of somatic cells provides benefits to the organism

The composition effect profiles $F_{\text{comp}}(x)$ that promote the evolution of irreversible somatic differentiation have certain characteristic shapes, see *Figure 2E–H*. We investigated what kind of composition effect profiles can make irreversible somatic differentiation become an evolutionary optimum. We sampled a number of random composition effect profiles with independently drawn parameter values and found optimal developmental strategies for each profile for a number of differentiation costs ($c$) and maturity size ($2^n$) values. We took a closer look at the instances of $F_{\text{comp}}(x)$ which resulted in irreversible somatic differentiation being evolutionarily optimal.

We found that irreversible strategies are only able to evolve when the soma-role cells contribute to the organism cell doubling time even if present in small proportions, see *Figure 3A,B*. Analysing parameters of the composition factors promoting irreversible differentiation, we found that this effect manifests in two patterns. First, the contribution threshold value ($x_0$) has to be small, see *Figure 3D* – irreversible differentiation is promoted when soma-role cells begin to contribute to the organism growth even in low numbers. Second, the contribution synergy was found to be large ($\alpha > 1$) or, alternatively, the saturation threshold ($x_1$) was small, see *Figure 3C*.

Both the contribution threshold $x_0$ and the contribution synergy $\alpha$ control the shape of the composition effect profile at intermediary abundances of soma-role cells. If the contribution synergy $\alpha$ exceeds 1, the profile is convex, so the contribution of soma-role cells quickly becomes close to maximum benefit ($b$). A small saturation threshold ($x_1$) means that the maximal benefit of soma is achieved already at low concentrations of soma-role cells (and then the shape of composition effect profile between two close thresholds has no significance). Together, these patterns give an evidence that the most crucial factor promoting irreversible somatic differentiation is the effectiveness of soma-role cells at small numbers, see Appendix 4 for more detailed data presentation.

These patterns are driven by the static character of differentiation strategies we use: the chances for a cell to differentiate are the same at the first and the last round of cell division. Therefore, the optimal germ-to-soma differentiation rate is found as a balance between the needs to deploy soma-role cells early on and to keep the high number of germ-role by the end of the life cycle. This implies that irreversible somatic differentiation strategies produce soma-role cells at lower rate than reversible strategies, see *Figure 2D*. With irreversible differentiation, an organism spends a significant amount of time having only a few soma-role cells. Hence, the irreversible strategy can only be evolutionarily successful, if the few soma-role cells have a notable contribution to the organism growth time.

We also found that profiles featuring irreversible differentiation do not possess neither extremely large, nor extremely small maximal benefit values $b$, see *Figure 3D*. When the maximal benefit is too small, the cell differentiation just does not provide enough benefits to be selected for and the evolutionarily optimal strategy is no differentiation. In the opposite case, when the maximal benefit is very close to one, the cell doubling time approaches zero, see *Equation (3)*. Then, the benefits of having many soma-role cells outweighs the costs of differentiation and the optimal strategy is reversible, see Appendix 4.

## For irreversible somatic differentiation to evolve, the organism size must be large enough

By screening through the maturity size ($2^n$) and differentiation costs ($c$), we found that the evolution of irreversible somatic differentiation is heavily suppressed at small maturity sizes, *Figure 4A*. We found that either reversible strategies or the no differentiation strategy evolve in small organisms. Since reversible strategies can quickly reach a fixed fraction of soma-role cells, thus they can obtain maximised benefits from soma-role cells with small maturity sizes (*Appendix 2—figure 1*). Since the no differentiation strategy does not involve cell differentiation, they do not have cell differentiation costs. In contrast, irreversible strategies increase the fraction of soma-roles and increase the benefits of soma-role cells gradually as maturity size increases. Meanwhile, the cell differentiation costs for irreversible strategies decrease as maturity size increases as the fraction of germ-role cells decreases.

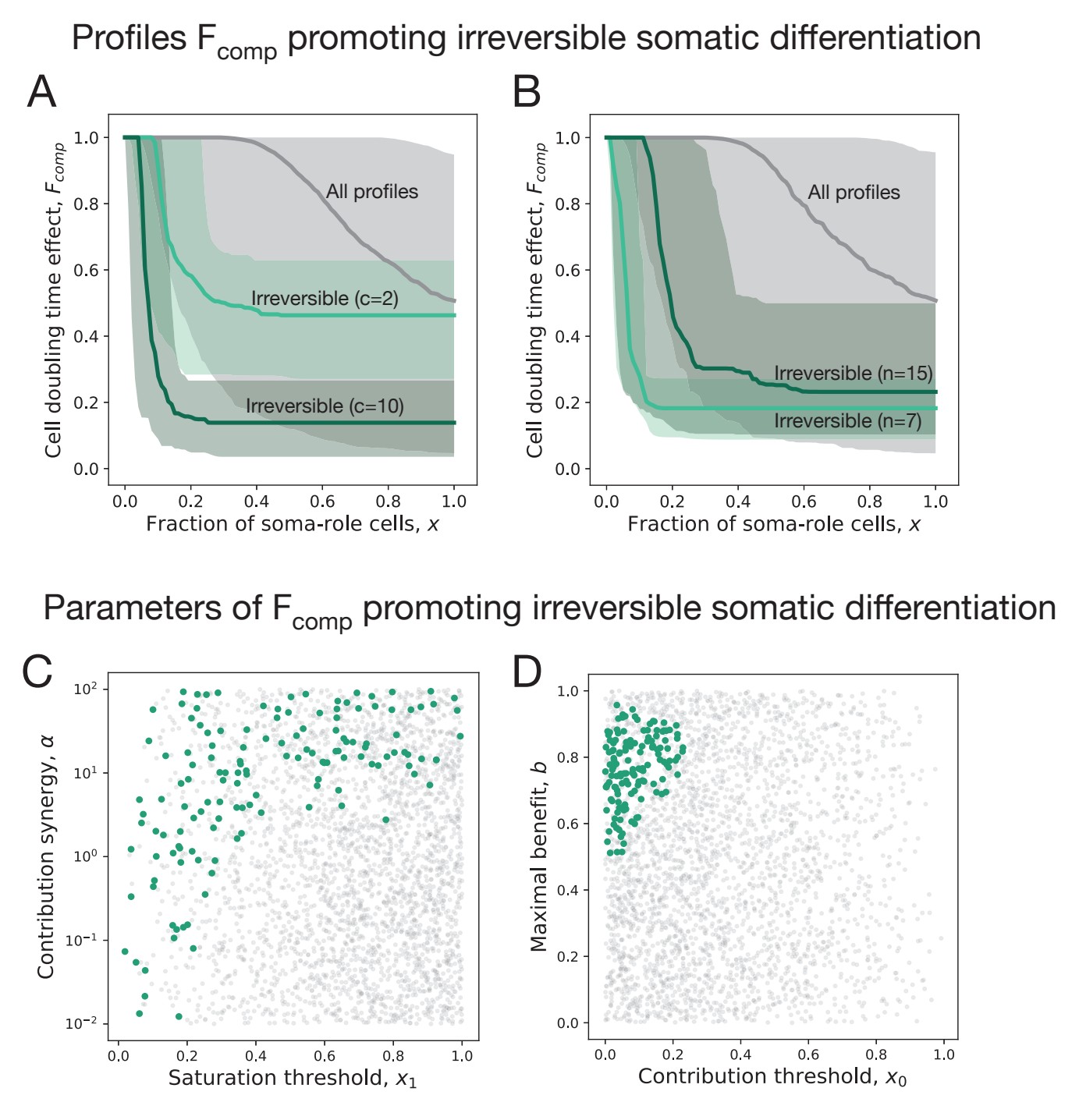

**Figure 3.** Irreversible soma evolves when substantial benefits arise at small concentrations of soma-role cells. In all panels, the data representing the entire set of composition effect profiles $F_{comp}(x)$ is presented in grey, while the subset promoting irreversible strategies is coloured. (**A, B**) Median and 90% confidence intervals of composition effect profiles at different differentiation costs (A, number of cell division $n = 10$) and maturity sizes (B, differentiation costs $c = 5$). (**C, D**) The set of composition effect profiles in the parameter space. Each point represents a single profile ($c = 5$ and $n = 10$). (**C**) The co-distribution of the saturation threshold ($x_1$) and the contribution synergy ($\alpha$) reveals that either $x_1$ must be small or $\alpha$ must be large. (**D**) Co-distribution of the contribution threshold ($x_0$) and the maximal benefit ($b$) shows that $x_0$ must be small, while $b$ must be intermediate to promote irreversible differentiation. A total of 3000 profiles are used for panels A, C, D and 1000 profiles for panel B.

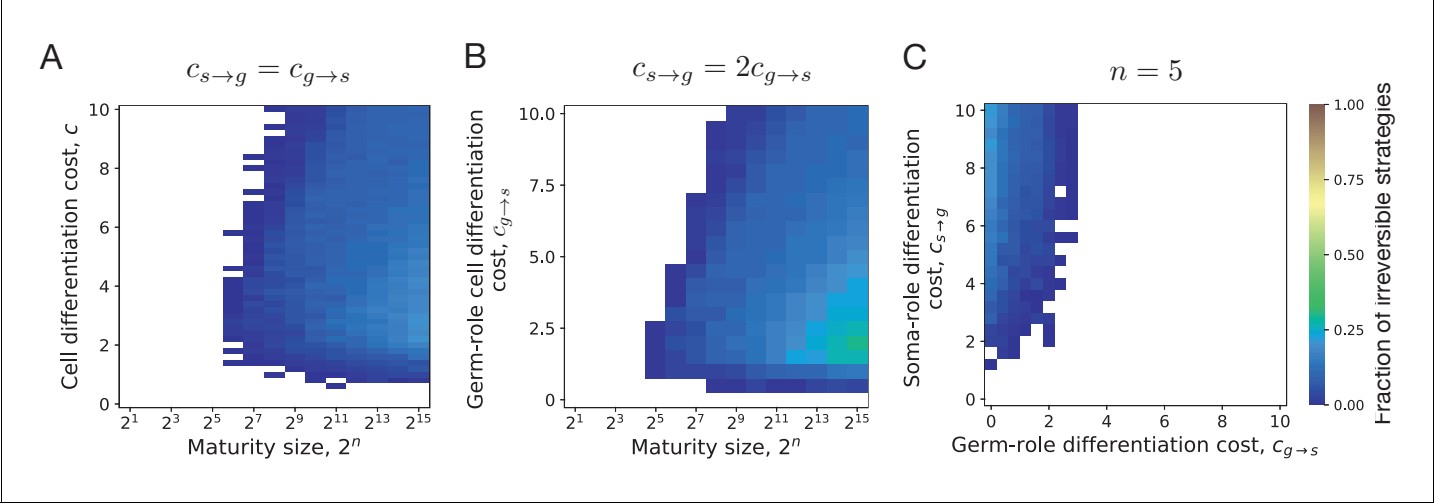

**Figure 4.** Irreversible differentiation can evolve if organism grows to a large enough size in the course of its life cycle. (**A**) The fraction of composition effect profiles promoting irreversible strategies at various cell differentiation costs ($c = c_{s \to g} = c_{g \to s}$) and maturity sizes ($2^n$). Irreversible strategies were only found for maturity size $2^6 = 64$ cells and larger. (**B**) The fraction of composition effect profiles promoting irreversible strategies at unequal differentiation costs $c_{s \to g} = 2c_{g \to s}$. A rare occurrences of irreversible strategies (~1%) was detected at the maturity size $2^5 = 32$ cells in a narrow range of cell differentiation costs but not at the smaller sizes. (**C**) The range of cell differentiation costs allowing evolution of irreversible strategies at at the maturity size $2^n = 32$ ($n = 5$) cells. For irreversible strategies to evolve at such a small size, the differentiation from soma-role to germ-role must be much more costly than the opposite transition ($c_{s \to g} \gg c_{g \to s}$).

Thus compared with other strategies, the irreversible strategies have advantages in large organisms. We found that under $c_{s \to g} = c_{g \to s}$, the minimal maturity size allowing irreversible somatic differentiation to evolve is $2^n = 64$ cells. At the same time, organisms performing just a few more rounds of cell divisions are able to evolve irreversible differentiation at a wide range of cell differentiation costs, see also Appendix 5. This indicates that the evolution of irreversible somatic differentiation is strongly tied to the size of the organism.

Evolution of irreversible strategies at sizes smaller than 64 cells is possible for $c_{s \to g} > c_{g \to s}$. For instance, at $c_{s \to g} = 2c_{g \to s}$ some irreversible strategies were found to be optimal at the maturity size $2^5 = 32$ cells, *Figure 4B*. However, irreversible strategies were found in a narrow range of cell differentiation costs and the fraction of composition effect profiles that allow evolution of irreversible differentiation there was quite low – about 1%. The evolution of irreversible strategies at such small maturity sizes becomes likely only at extremely unequal costs of transition between germ and some roles $c_{s \to g} \gg c_{g \to s}$, see *Figure 4C*. Hence, for irreversible somatic differentiation to evolve, the organism size should exceed a threshold of roughly 64 cells.

## Irreversible somatic differentiation can also evolve when cell differentiation is risky

In our main model, we considered differentiation costs in a specific form of cell division delay. However, the process of cell differentiation may impact the organism development in another way. Differentiation requires modifications in DNA regulation, which in turn poses a risk of dysregulation resulting in an emergence of selfish mutants that could for example cause cancer. The disposable soma theory suggests that cells performing vegetative functions form separate lineages to contain emerging mutations and prevent them from passing to the next generations of organisms. In line with this hypothesis, we also considered a model of risky cell differentiation, where the transition between germ and soma roles incurs a risk of getting cancer that kills the entire organism, see Appendix 6.

The results obtained with a model of risky differentiation are very similar to the outcomes of our main model, where cell differentiation cause delay, see *Figure 5*. In both models, irreversible differentiation only evolves if cell differentiation does not come for free but brings costly side-effects (delay or risk). Also, in both models irreversible differentiation is prevalent when costs of soma-to-

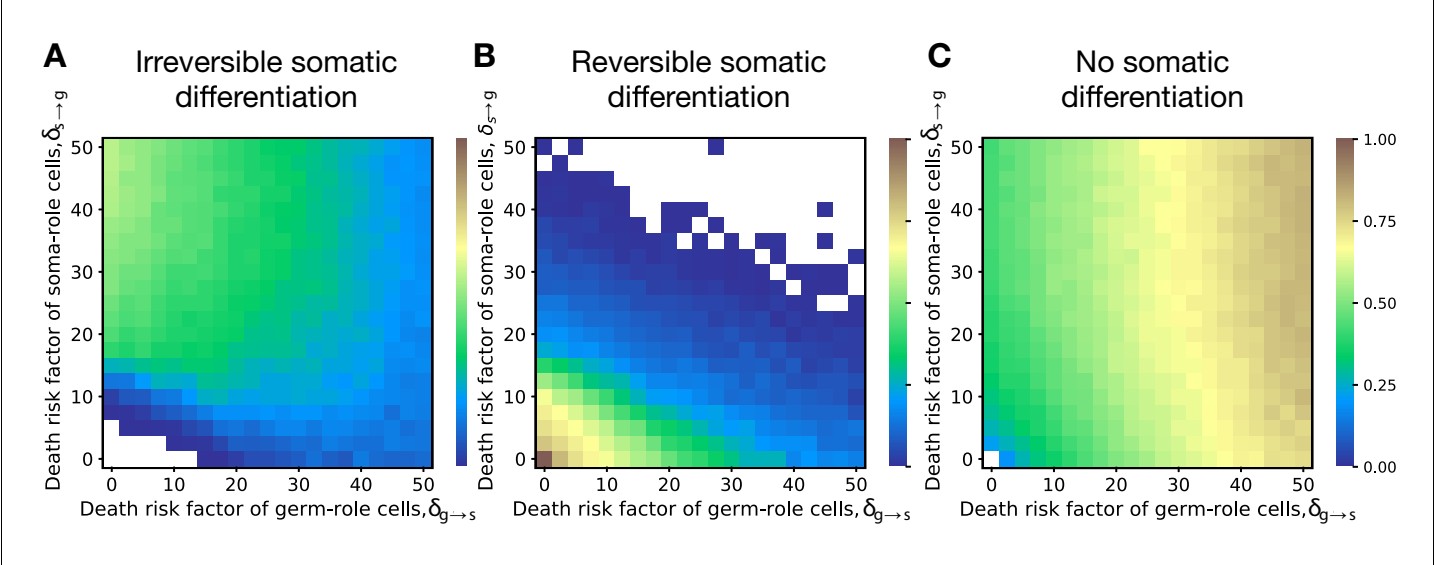

**Figure 5.** Irreversible differentiation can evolve if cell differentiation incurs a risk of organism death instead of division delay. The fractions of 200 random composition effect profiles promoting irreversible (**A**), reversible (**B**), and no differentiation (**C**) strategies at various risks of cell differentiation ($\delta_{s \to g}$, $\delta_{g \to s}$). The qualitative pattern is very similar to the results obtained with the model of cell differentiation causing delay, see *Figure 2A–C*. In the absence of risk ($\delta_{g \to s} = \delta_{s \to g} = 0$), only reversible strategies were observed. Reversible strategies are prevalent at smaller risk values. No differentiation strategies are the most abundant at large risk of germ-to-soma transition ($\delta_{g \to s}$). Irreversible strategies are the most abundant at large risks of soma-to-germ transition ($\delta_{s \to g}$). The maturity size used in the calculation is $2^{10}$ cells. The legend is the same as that in *Figure 2A–C*.

germ transitions are intense; reversible differentiation is prevalent when costs of both transitions are low; and no differentiation is prevalent when costs of germ-to-soma transitions are intense *Figure 2A–C*.

## Discussion

The vast majority of cells in a body of any multicellular being contains enough genetic information to build an entire new organism. However, in a typical metazoan species, very few cells actually participate in the organism reproduction – only a limited number of germ cells are capable of doing it. The other cells, called somatic cells, perform vegetative functions but do not contribute to reproduction – somatic differentiation is irreversible. We asked for the reason for the success of such a specific mode of organism development. We theoretically investigated the evolution of irreversible somatic differentiation with a model of clonally developing organisms taking into account benefits provided by soma-role cells, costs arising from cell differentiation, and the effect of the raw organism size.

Our key findings are:

- The evolution of irreversible somatic differentiation is inseparable from costly cell differentiation or risky cell differentiation.
- For irreversible somatic differentiation to evolve in organisms with synchronous cell division, somatic cells should be able to contribute to the organism performance already when their numbers are small.
- Only large enough organisms tend to develop irreversible somatic differentiation.

According to our results, cell differentiation costs are essential for the emergence of irreversible somatic differentiation, see *Figure 2A*. The costs punish strategies with high rate of cell differentiation. As a result, irreversible strategies gain an advantage because their overall differentiation rate is low, see *Figure 2D*, and soma-role cells do not differentiate at all. Most models focus on traits that lead to benefits for the organism, while the cost of cell differentiation are rarely considered. For cells in a multicellular organism, differentiation costs arise from the material needs, energy, and time it takes to produce components necessary for the performance of the differentiated cell, which were

absent in the parent cell. For instance, in filamentous cyanobacteria nitrogen-fixating heterocysts develop much thicker cell wall than parent photosynthetic cells had. Also, reports indicate between 23% (*Ow et al., 2008*) and 74% (*Sandh et al., 2014*) of the proteome changes its abundance in heterocysts compared against photosynthetic cells. Similarly, the changes in the protein composition in the course of cell differentiation was found during the development of stalk and fruiting bodies of *Dictyostelium discoideum* (*Bakthavatsalam and Gomer, 2010*; *Czarna et al., 2010*).

An alternative to differentiation costs in terms of slower growth is a model with a risky differentiation, where we found similar patterns, see *Figure 5*. These results indicate that the exact mechanism of the differentiation costs does not play a major role in the evolution of irreversible somatic differentiation.

Our model demonstrates that irreversible somatic differentiation is more likely to evolve when a few soma-role cells are able to provide a substantial benefit to the organism, see *Figure 3*. *Volvocales* algae demonstrate that a significant contribution by small numbers of somatic cells might indeed be found in a natural population: In *Eudorina illinoiensis*, only four out of thirty-two cells are vegetative (*Sambamurty AVSS, 2005*) (soma-role in our terms). This species has developed some reproductive division of labour and a fraction of only $1/8$ of vegetative cells is sufficient for colony success. Thus, it seems possible that highly-efficient soma-role cells open the way to the evolution of irreversible somatic differentiation. Several patterns of how cells proved the benefit to an organism have been previously considered (*Michod, 2007*; *Willensdorfer, 2009*; *Rossetti et al., 2010*; *Rueffler et al., 2012*; *Cooper and West, 2018*; *Yanni et al., 2020*). The majority of papers focuses on the resource allocation toward different tasks in each cell in an organism and how divergent different cells can be. In our model, we assume that the germ-role and soma-role cell are different in function and focus on the relationship between the number of soma-role cells and their impact, e.g. the character of their interactions. While the found $F_{\mathrm{comp}}$ curves exhibit convex-like shape, see *Figure 3A,B*, this finding has a different nature from the convex trade-off between fertility and viability found in the models of cell differentiation (*Michod, 2007*).

Our model shows that irreversible somatic differentiation does not evolve if the organism size is small, see *Figure 4A*. The maturity size plays an important role in an organism's life cycle (*Amado et al., 2018*; *Erten and Kokko, 2020*): Large organisms have potential advantages to optimize themselves in multiple ways, such as to improve growth efficiency (*Waters et al., 2010*), to avoid predators (*Matz and Kjelleberg, 2005*; *Fisher et al., 2016*; *Hiltunen and Becks, 2014*), to increase problem-solving efficiency (*Morand-Ferron and Quinn, 2011*), and to exploit the division of labour in organisms (*Carroll, 2001*; *Matt and Umen, 2016*). Moreover, the maximum size has been related to the reproduction of the organism from the onset of multicellularity in Earth's history (*Ratcliff et al., 2012*). Our results suggest that the smallest organism able to evolve irreversible somatic differentiation should typically be about 32–64 cells (unless the cost of soma-to-germ differentiation is extremely large and the cost of the reverse is low). This is in line with the pattern of development observed in *Volvocales* green algae. In *Volvocales*, cells are unable to move (vegetative function) and divide (reproductive function) simultaneously, as a unique set of centrioles are involved in both tasks (*Wynne and Bold, 1985*; *Koufopanou, 1994*). *Chlamydomonas reinhardtii* (unicellular) and *Gonium pectorale* (small colonies up to 16 cells) perform these tasks at different times. They move towards the top layers of water during the day to get more sunlight. At night, however, these species perform cell division and/or colony reproduction, slowly sinking down in the process. However, among larger *Volvocales*, a division of labour begins to develop. In *Eudorina elegans* colonies, containing 16–32 cells, a few cells at the pole have their chances to give rise to an offspring colony reduced (*Marchant, 1977*; *Hallmann, 2011*). In *P. californica*, half of the 128-celled colony is formed of smaller cells, which are totally dedicated to the colony movement and die at the end of colony life cycle (*Kikuchi, 1978*; *Hallmann, 2011*). In *Volvox carteri*, most of a 10,000 cell colony is formed by somatic cells, which die upon the release of offspring groups (*Hallmann, 2011*).

In a majority of our tests, we used the maturity size of $2^{10} = 1024$ cells. This is significantly larger than the minimal necessary size for evolution of irreversible somatic differentiation. However, the body size of the order of 1000 cell attracts attention because at this scale organisms of very diverse degrees of complexity are observed: from undifferentiated colonies (ocean algae *Phaeocystis antarctica*), to intermediary life forms (slime molds slugs), to paradigm multicellular organisms (higher *Volvocales* and nematode *Caenorhabditis elegans*).

The model presented in our study focuses on the transition from colonial life forms to multicellular beings. Further development of complexity opens multiple new ways for optimization of life cycle. For example, a maternal organism can provide protection and nurture for offspring at their early stages of growth, like in *V. carteri* (10,000 cells) in which offspring colonies develop inside the parental organism. There, the rate of offspring growth depends mostly on the performance of the maternal organism and much less on the differentiation strategy of offspring. Having maternal protection allows to relax the conditions for evolution of irreversible differentiation indicated in our study. How much these conditions can be relaxed is a very interesting question.

One of the most significant assumptions we took is the synchronicity of cell divisions even if division outcomes are different. This is only possible if cell actions are coordinated at the level of organism – otherwise, cells that do not differentiate may complete their divisions before differentiating cells. When in the history of multicellularity such a coordination emerges is an open question. However, in a number of rather simple species, a synchronicity of cell divisions paired with cell differentiation is observed. One example is the green algae *Eudorina illinoiensis* – one of the simplest species demonstrating the first signs of reproductive division of labour, in which four out of 32 cells are differentiated (*Sambamurty AVSS, 2005*). Another example is 128-celled algae *Pleodorina californica*, half of the cells are differentiated. And still, the cell divisions are synchronous (*Kikuchi, 1978*). Even the size of the mature organism being a power of two indicates that cells do not divide independently, but their actions are controlled at the level of the organism.

To peek at the impact of the cell division synchronicity, we developed a model with asynchronous cell division, where cell differentiation costs are paid individually by each differentiating cell, see Appendix. G. We found that the evolution of irreversible differentiation is significantly suppressed even under the most favourable conditions ($c_{s \to g} \gg c_{g \to s}$) – the frequency of composition profiles promoting irreversible somatic differentiation is much smaller and the maturity size restriction is higher.

Another assumption, which shapes the results of our study, is the static differentiation strategy the probability of each division outcome does not depend on the stage of life cycle. On the one hand, the static nature of differentiation strategy puts irreversible differentiation in disadvantage, as it creates a trade-off between the fraction of soma-role cells at the early stage of life cycle and the number of germ-role cells at the end of life cycle. On the other hand, a set of fully flexible dynamic differentiation strategies present an efficient but hardly realistic solution to the life cycle optimization problem: at the first round of cell divisions organism converts to all-soma state and remains so until the last round, when all cells convert back to germ-state. Theoretically, this strategy provides simultaneously the fastest possible development rate (100% soma-role cells during life cycle) and the largest possible number of offspring (100% germ-role cells at the end of life cycle). Still, we cannot provide an example of such a developmental program in nature. Nevertheless, the differentiation strategy of higher *Volvocales* is not static *Kirk, 2005* and the exploration of a vast space of dynamic differentiation strategies warrants further investigation.

We acknowledge that our discussion of natural examples of germ-soma differentiation relies heavily on *Volvocales* algae. This merely reflects the bias in the empirical literature about evolution of germ/soma differentiation towards this group. We should note that our model is not a model of *Volvocales* life cycle. Instead, we aim to answer the question about emergence of irreversible somatic differentiation in a broad context without tailoring it to the features of a single group.

Our study originated from curiosity about driving factors in the evolution of irreversible somatic differentiation: Why does the green algae *Volvox* from the kingdom Plantae shed most of its biomass in a single act of reproduction? And why, in another kingdom, Animalia, in most of the species the majority of body cells is outright forbidden to contribute to the next generation? Our results show which factors makes a difference between the evolution of an irreversible somatic differentiation and other strategies of development. One of these factors, the maturity size, is known in the context of the evolution of reproductive division of labour (*Kirk, 2005*). Another factor, the costs of cell differentiation, is, in general, discussed in a greater biological scope but is hardly acknowledged as a factor contributing to the evolution of organism development. Finally, the early contribution of soma-role cells to the organism growth, even if they are small in numbers, is an unexpected outcome of our investigation, overlooked so far as well. Despite the simplistic nature of our model (we did not aim to model any specific organism), all our results find a confirmation among the *Volvocales* clade. Hence, we expect that the findings of this study reveal general properties of the evolution of irreversible somatic differentiation, independently of the clade where it evolves.

## Acknowledgements

YG, HJP, AT, and YP thank the Max Planck Society for generous funding. HJP was supported by the NRF grant funded by the Korea government (MSIT) Grant No.2020R1A2C1101894 and by an appointment to the JRG Program at the APCTP through the Science and Technology Promotion Fund and Lottery Fund of the Korean Government. This was also supported by the Korean Local Governments - Gyeongsangbuk-do Province and Pohang City.

## Additional information

### Funding

| Funder | Grant reference number | Author |
| --- | --- | --- |
| Ministry of Science and ICT, South Korea | 2020R1A2C1101894 | Hye Jin Park |
| JRG Program | | Hye Jin Park |

The funders had no role in study design, data collection and interpretation, or the decision to submit the work for publication.

### Author contributions

Yuanxiao Gao, Conceptualization, Software, Formal analysis, Investigation, Visualization; Hye Jin Park, Conceptualization, Software, Methodology; Arne Traulsen, Conceptualization, Supervision; Yuriy Pichugin, Conceptualization, Methodology, Project administration

### Author ORCIDs

Yuanxiao Gao (ID) https://orcid.org/0000-0002-6719-8299
Hye Jin Park (ID) https://orcid.org/0000-0003-3552-6275
Arne Traulsen (ID) https://orcid.org/0000-0002-0669-5267
Yuriy Pichugin (ID) https://orcid.org/0000-0003-3078-2499

### Decision letter and Author response

Decision letter https://doi.org/10.7554/eLife.66711.sa1
Author response https://doi.org/10.7554/eLife.66711.sa2

## Additional files

### Supplementary files

• Transparent reporting form

### Data availability

The code implementing our model is deposited at https://github.com/YuanxiaoGao/Evolution-of-irreversible-somatic-differentiation (copy archived at https://archive.softwareheritage.org/swh:1:rev:9a1ea7c84f3041ebe3720e7837b28182912b5e00).

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

## Appendix 1

### Search for the evolutionarily optimal developmental program

Finding the population growth rate for a given developmental program

In *Gao et al., 2019*, we have shown that a population of organisms, which begin their life cycle from the same state but have a stochastic development, eventually grows exponentially with the rate $\lambda$ given by the solution of

$$\sum_i e^{-\lambda T_i} G_i P_i = 1. \tag{5}$$

Here, $i$ is the developmental trajectory – in our case, the specific combination of all cell division outcomes; $P_i$ is the probability that an organism development will follow the trajectory $i$; $T_i$ is the time necessary to complete the trajectory $i$ – from a single cell to the maturity size of $2^n$ cells; $G_i$ is the number of offspring organisms produced at the end of developmental trajectory $i$, equal to the number of germ-role cells at the moment of maturity.

In order to find the population growth rate, we need to know $G_i$, $T_i$, and $P_i$ (how many offspring are produced, how long did it take to mature, and how likely is this developmental trajectory, respectively). The complete set of developmental trajectories is huge as it scales exponentially with the number of divisions $n$.

In our study, for each developmental strategy, we sampled $M = 300$ developmental trajectories at random. To get each trajectory, we simulated the growth of the single organism according to the rules of our model. For each trajectory, the developmental time $T_i$ was computed as a sum of cell doubling times at each of the $n$ synchronous cell divisions, the number of offspring $G_i$ was given by the count of germ-role cells at the end of development. The resulting ensemble of trajectories (with $P_i = 1/M$) was plugged into (5) to compute the population growth rate $\lambda$.

### Finding the developmental program with the largest population growth rate

We assume that evolution occurs by growth competition between populations executing different developmental strategies. These strategies, which provide larger population growth rate will out-grow others. To find evolutionarily optimal strategies under given conditions, we screened through a large set of developmental strategies and identified the one with the maximal population growth rate $\lambda$. Since the probabilities of cell division outcomes sum into one ($g_{gg} + g_{gs} + g_{ss} = 1$ and $s_{gg} + s_{gs} + s_{ss} = 1$), these probabilities can be represented as a point on two simplexes, one for the division of germ-role cells, and one for the division of soma-role cells. Consequently, we choose the set of developmental strategies as a Cartesian product of two triangular lattices – one for division probabilities of germ-role cells ($g_{gg}, g_{gs}, g_{ss}$) and one for soma-role cells ($s_{gg}, s_{gs}, s_{ss}$). The lattice space was set to 0.1, so each of two independent lattices contained $11 \times 12/2 = 66$ nodes, and the whole set of developmental strategies comprised $66 \times 66 = 4356$ different strategies. For each of these strategies, the population growth rate $\lambda$ was calculated and the strategy with the largest growth rate was identified as evolutionarily optimal.

In our investigation, parameters such as differentiation costs ($c_{s \to g}$, $c_{g \to s}$) and maturity size ($2^n$) were used as control parameters. In other words, we either fix them at the specific values, or screened through a range of values to obtain a map (see *Figures 2* and *3* in the main text). However, the parameters that controlled the shape of composition effect profile ($x_0$, $x_1$, $\alpha$, and $b$) were treated differently. For each combination of control parameters, we randomly sampled a number (between 200 and 3000) of combinations of these parameters. The thresholds ($0 \le x_0 \le x_1 \le 1$) were sampled as a pair of independent distributed random values from the uniform distribution $U(0, 1)$. The contribution threshold $x_0$ was set to the minimum of the pair, and the saturation threshold $x_1$ was set to the maximum. The contribution synergy ($\alpha > 0$) corresponds to the concave shape of the profile at $\alpha < 1$ and to the convex shape at $\alpha > 1$. Therefore, $\log_{10}(\alpha)$ was sampled from the uniform distribution $U(-2, +2)$, so the profile has an equal probability to demonstrate concave and convex shape. Finally, the maximum benefit ($0 \le b < 1$) was sampled from a uniform distribution, $U(0, 1)$. For each tested combination of control parameters, we found the optimal developmental strategy for every sampled

profile. We then classified these as irreversible somatic differentiation, reversible somatic differentiation, or no somatic differentiation.

## Appendix 2

### Under costless cell differentiation, irreversible soma strategy cannot be evolutionarily optimal

In this section, we will show that an irreversible strategy can never be an evolutionary optimum without cell differentiation being costly. To do that, we first consider the deterministic dynamics of the expected composition of the organism. Then, for an arbitrary irreversible strategy, we identify a more advantageous reversible strategy which gives the same organism composition at the end of life cycle but higher number of soma-role cells during the life cycle.

In our model, the composition of the organism is governed by the stochastic developmental strategy and differs between different organisms. Here, as a proxy for this complex stochastic dynamics, we consider the mathematical expectation of the composition. Assume that after $j$ cell divisions the fraction of soma-role cells is $r_s(j)$ and the fraction of germ-role cells is $r_g(j) = 1 - r_s(j)$ , $j = 1, \ldots, n$, where $n$ is the maximal number of divisions. Then, the expected fractions of cells of the two types after the next cell division is

$$r_s(j+1) = \left(s_{ss} + \frac{s_{gs}}{2}\right)r_s(j) + \left(\frac{g_{gs}}{2} + g_{ss}\right)r_g(j) = (1 - m_s)r_s(j) + m_g r_g(j),$$

$$r_g(j+1) = \left(g_{gg} + \frac{g_{gs}}{2}\right)r_g(j) + \left(\frac{s_{gs}}{2} + s_{gg}\right)r_s(j) = (1 - m_g)r_g(j) + m_s r_s(j) \tag{6}$$

where we introduced $m_s = s_{gg} + \frac{s_{gs}}{2}$ and $m_g = g_{ss} + \frac{g_{gs}}{2}$ – the probabilities that the offspring of a cell will have a different role. Naturally, for irreversible somatic differentiation $m_s = 0$ and $m_g > 0$ , for no somatic differentiation strategies $m_g = 0$ and $m_s$ being irrelevant, while the reversible differentiation class covers the rest. (6) can be written in matrix form

$$\begin{pmatrix} r_s(j+1) \\ r_g(j+1) \end{pmatrix} = \begin{pmatrix} 1 - m_s & m_g \\ m_s & 1 - m_g \end{pmatrix} \cdot \begin{pmatrix} r_s(j) \\ r_g(j) \end{pmatrix} \tag{7}$$

A newborn organism contains a single germ-role cell $(r_s(0) = 0, r_g(0) = 1)$ , therefore, the expected composition of an organism after $j$ divisions is

$$\begin{pmatrix} r_s(j) \\ r_g(j) \end{pmatrix} = \begin{pmatrix} 1 - m_s & m_g \\ m_s & 1 - m_g \end{pmatrix}^j \cdot \begin{pmatrix} 0 \\ 1 \end{pmatrix} \tag{8}$$

The matrix has two eigenvalues: 1 and $1 - m_g - m_s$, with associated right eigenvectors $(m_g, m_s)^T$ and $(1, -1)^T$, respectively. Hence, the expected composition after $j$ divisions can be obtained in the explicit form

$$r_s(j) = \frac{1}{m_g + m_s}\left[m_g - m_g(1 - m_g - m_s)^j\right],$$

$$r_g(j) = \frac{1}{m_g + m_s}\left[m_s + m_g(1 - m_g - m_s)^j\right]. \tag{9}$$

For an arbitrary irreversible somatic differentiation strategy $D$, $m_s = 0$, the expected number of soma-role cells changes as

$$r_{s,D}(j) = 1 - (1 - m_g)^j, \tag{10}$$

which is a monotonically increasing function of the number of cell divisions $t$, see the green line in Fig. B. In the life cycle involving $j$ cell divisions, the fraction of soma-role cells at the end of life cycle is $r_{s,D}(j) = 1 - (1 - m_g)^j$.

Now, we consider another developmental strategy $D'$ with reversible somatic differentiation in which $m'_g = r_{s,D}(n)$ and $m'_s = 1 - r_{s,D}(n)$. Using $m'_g + m'_s = 1$ in (9), it can be shown that the expected fraction of soma-role cells in $D'$ after the very first cell division is exactly $r_{s,D}(n)$ and stays constant

thereafter, see the orange line in Fig. B. Thus, the number of offspring produced is the same for both development strategies.

If cell differentiation is costless ($d_s = d_g = 0$), then the cell doubling time depends only on the fraction of soma-role cells. As all soma-role cells are then present already after the first cell division, organisms following the reversible strategy $D'$ will grow faster than organisms using the irreversible strategy $D$ at any stage of organism development, independently of the choice of the composition effect profile ($F_{\text{comp}}$). At the end of the life cycle, both strategies have the same expected number of offspring. Therefore, under costless cell differentiation, for any irreversible strategy, we can find a reversible strategy that leads to a larger population growth rate.

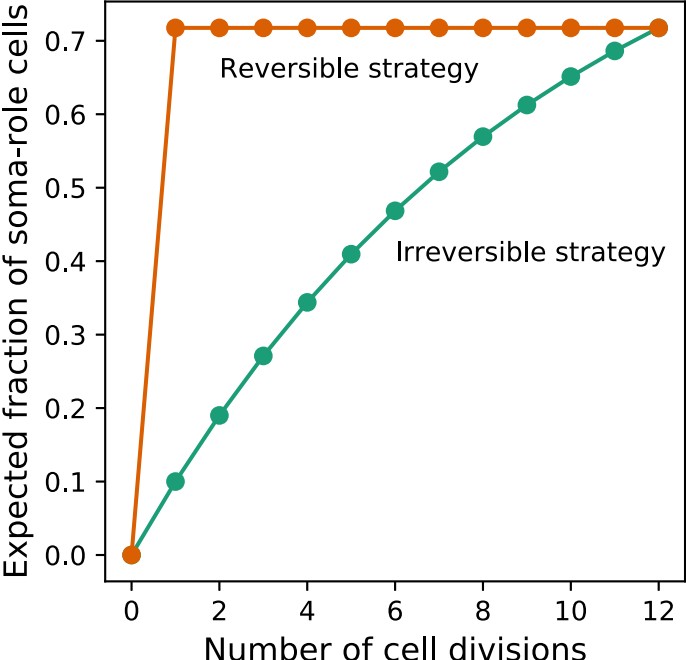

**Appendix 2—figure 1.** Under costless differentiation, for any irreversible somatic differentiation strategy, exists a reversible somatic differentiation strategy dominating it. The green curve shows the dynamics of the expected fraction of soma-role cells in an organism using an irreversible strategy ($m_g = 0.1$, $m_s = 0.0$, $n = 12$). The orange curve shows the dynamics of the expected fraction of soma-role cells in an organism using the specific reversible strategy [$m'_g = 1 - (1 - m_g)^{12} \approx 0.72$, $m'_s = 1 - m'_g \approx 0.28$]. In this strategy, the number of offspring produced at the end of the life cycle is the same as in the considered irreversible strategy. At the same time, the fraction of soma-role cells during the life cycle is larger. Therefore, under costless differentiation, the presented reversible strategy is more effective than the considered irreversible strategy.

## Appendix 3

### Conditions promoting the evolution of reversible, irreversible, and no differentiation strategies

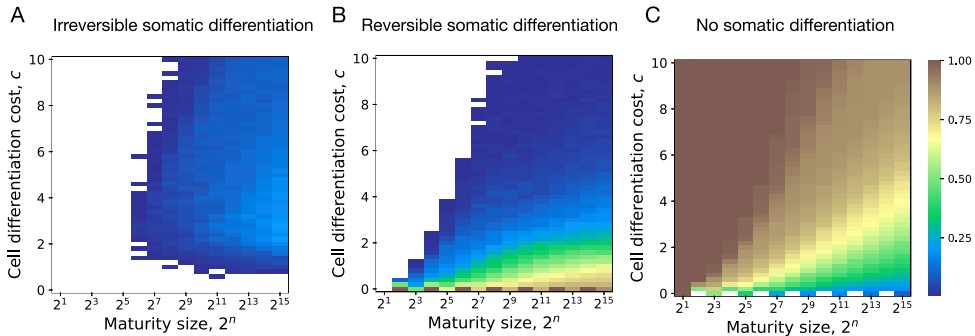

**Appendix 3—figure 1.** Impact of maturity size on the evolution of development strategies. The fractions of 200 random composition effect profiles promoting irreversible (**A**), reversible (**B**), and no differentiation (**C**) strategies at various cell differentiation costs ($c = c_{s \to g} = c_{g \to s}$) and maturity size $2^n$. Irreversible strategies are most abundant at large maturity sizes and intermediary cell differentiation costs. Reversible strategies are most abundant at small cell differentiation costs. No differentiation strategies are most abundant at small maturity sizes and cell differentiation costs. The legend is the same as that in *Figure 2A–C*.

## Appendix 4

## Parameters of composition effect profiles promoting reversible, irreversible, and no differentiation strategies

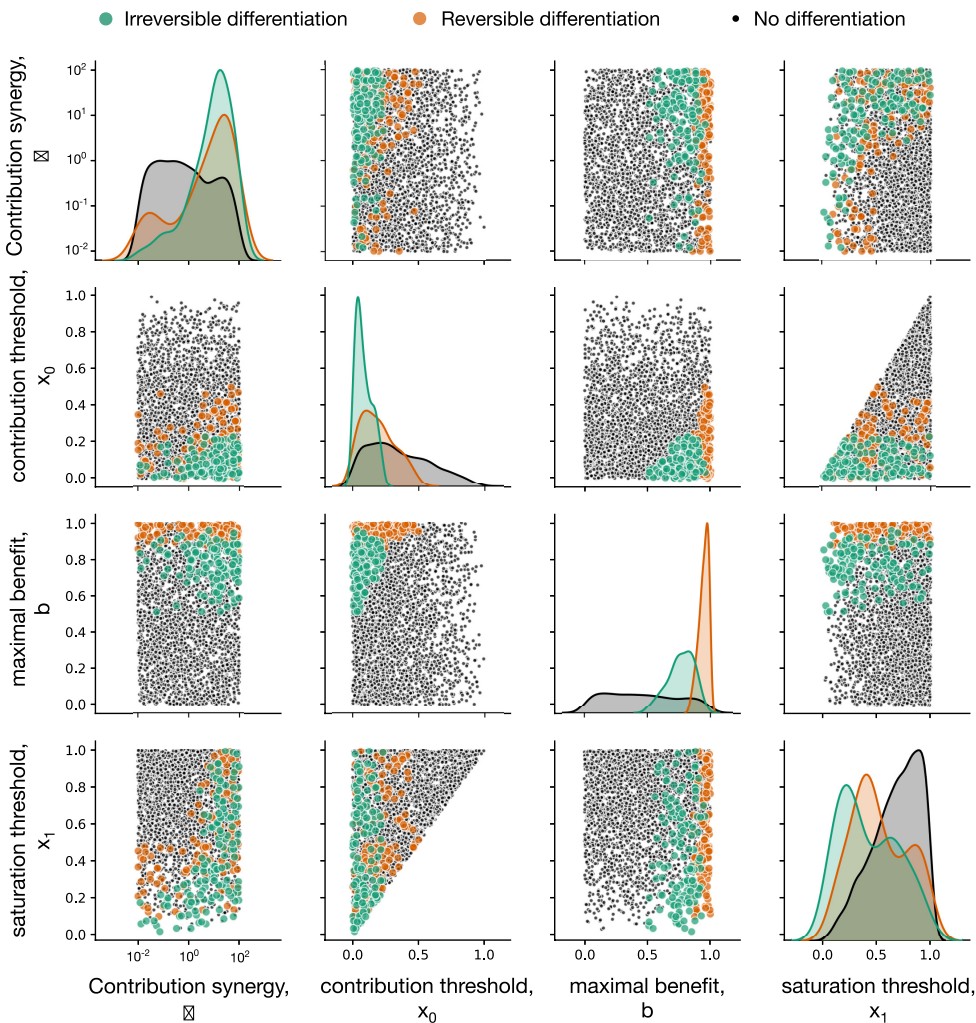

**Appendix 4—figure 1.** Impact of composition effect parameters on the evolution of development strategies. Each diagonal panel represents individual distribution of each of four parameters among composition effect profiles promoting irreversible (green), reversible (orange), and no differentiation (black) strategies. Each non-diagonal panel represents a pairwise co-distribution of these parameters. Irreversible (green) strategies are promoted at small contribution thresholds $x_0$ and intermediary maximal benefit $b$. Also, either the contribution synergy $\alpha$ must be large, or the saturation threshold $x_1$ should be small – see main text for detailed discussion. Reversible (orange) strategies require large $b$– there the benefits of having a large number of soma-role cells outweighs costs paid by frequent differentiation. Due to the fast accumulation of soma-role cells, reversible strategies tolerate larger $x_0$ than irreversible. Reversible exhibit the same restrictions with respect to $x_1$ as irreversible and are insensitive to $\alpha$. For this figure, 3000 composition effect profiles were investigated with costs $c = c_{s \to g} = c_{g \to s} = 5$ and $n = 10$.

## Appendix 5

### Evolution of irreversible somatic differentiation under various maturity sizes and unequal cell differentiation costs

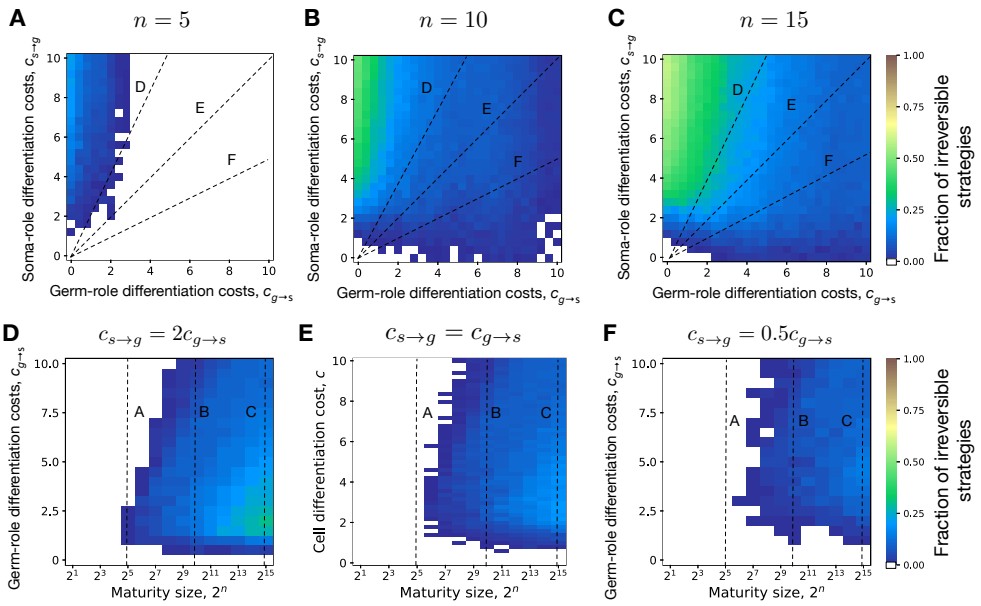

**Appendix 5—figure 1.** Evolution of irreversible somatic differentiation at unequal cell differentiation costs. (**A-C**) The fraction of 200 random composition effect profiles promoting irreversible strategy at various cell differentiation costs ($c_{s \to g}$, $c_{g \to s}$) at fixed maturity size $n = 5$ (panel A), 10 (**B**), and 15 (**C**). Larger maturity sizes promote the evolution of irreversible differentiation across all cell differentiation costs. (**D-F**) The fraction of composition effect profiles promoting irreversible strategy at unequal cell differentiation costs $c_{s \to g}/c_{g \to s} = 2$ (panel D), $c_{s \to g}/c_{g \to s} = 1$ (**E**), and $c_{s \to g}/c_{g \to s} = 0.5$ (**F**). Even with unequal differentiation costs, the minimal maturity size allowing the evolution of irreversible differentiation stays roughly the same — $2^5 - 2^6$ cells. Dashed lines indicate overlap between panels. The legend is the same as that in *Figure 2A–C*.

## Appendix 6

### Model of risky cell differentiation

In the risky differentiation model, we assume that cell differentiation implies a risk of errors leading to defective cells (*Aktipis et al., 2015*). These cells act in their selfish interests, compromising the integrity of an organism. This leads to the organism death, very similar to outcomes of cancer in complex multicellular species.

The impact of the defective cell depends on which stage of life cycle it appears. A defective cell emerged during the first cell division will likely result in a non-viable organism. At the same time, a defective cell emerged in the very last round of cell divisions is unlikely to affect the organism because its life cycle is about to end. To reflect this effect, we scaled the impact of a newly emerged defective cell by the number of cells already present in an organism. This way, the probability to get cancer is proportional to the frequency of cell differentiation events. The proportions of soma-role cells and germ-role cells that differentiate upon division in the total number of cell divisions are

$$f_{s \to g} = \frac{N_{s \to gg} + \frac{1}{2} N_{s \to gs}}{Z},$$

$$f_{g \to s} = \frac{N_{g \to ss} + \frac{1}{2} N_{g \to gs}}{Z} \tag{11}$$

where $N_{x \to yz}$ is the number of cell divisions at which cell of role $x$ gives rise to a $y$ cell and a $z$ cell, and $Z = 2^n - 1$ is the total number of cell divisions during the organism growth with maturity size $2^n$.

We define the probabilities of death caused by defective cells emerged in germ to soma and soma to germ transitions as

$$d_{g \to s} = \tanh(\delta_{g \to s} f_{g \to s}),$$

$$d_{s \to g} = \tanh(\delta_{s \to g} f_{s \to g}) \tag{12}$$

where $\delta_{g \to s}$ and $\delta_{s \to g}$ characterize the risk of cancer from a germ to soma and from a soma to germ transition. The transformation function $\tanh(x)$ is chosen to grow linearly at a small number of differentiation events but exponentially saturates to one if these events are numerous, see Fig. F.

We assume that an organism successfully completes its life cycle and produces offspring only if no cancer emerges in the course of its growth. The probability of this at each round of cell division is

$$P_{\text{success}} = (1 - d_{g \to s})(1 - d_{s \to g}). \tag{13}$$

Otherwise, the organism dies and does not produce any offspring. There are no delay differentiation costs in this model ($c_{s \to g} = c_{g \to s} = 0$).

A typical feature of the cancer cells in complex organisms is a high cell division rate. This has a large impact on organisms of complex animals, in which the division rate of regular cells is low and the life cycle are long. However, organisms in the focus of our study have very short life cycles (few rounds of cell divisions) and even the regular cells actively proliferate. Hence, the growth advantage of defective cells should have much smaller impact on simple species. Therefore, in this model, we neglect the difference in division rates between defective and regular cells and keep cell divisions synchronous.

The probability of getting cancer depends on the frequency of cell differentiation events. An organism with a higher cell differentiation rate has a higher death probability, which leads to slower population growth.

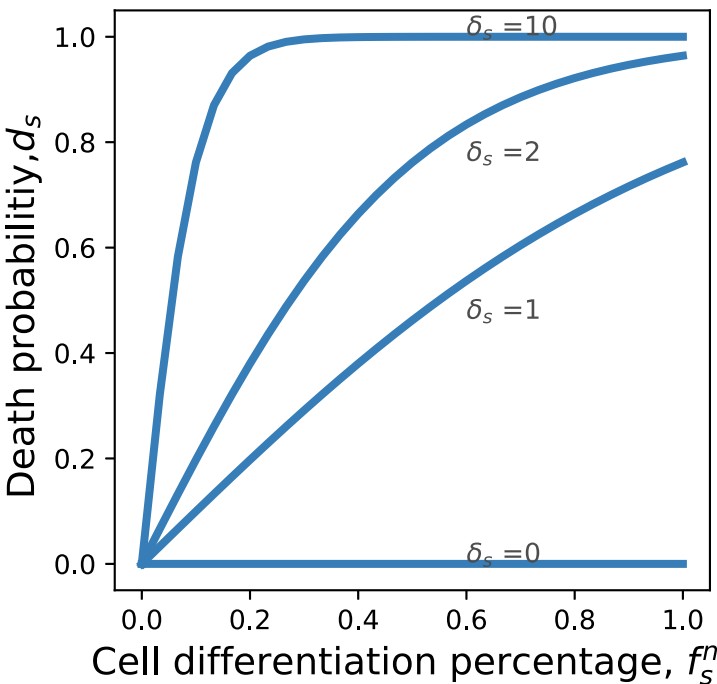

**Appendix 6—figure 1.** Probability of death from cancer as a function of the differentiation frequency. If the frequency of differentiation is small, the death probability grows linearly. If the frequency of differentiation is large, the death probability saturates at 1.

## Appendix 7

### Evolution of irreversible somatic differentiation in a model with asynchronous cell division

Our original model features synchronous cell division. This comes from the assumption that differentiation costs are paid collectively by the whole organism. Here, we consider another option, where differentiation costs are paid individually by each cell. An immediate consequence is that cell division in such a model is asynchronous because differentiating cells take more time to divide.

In the asynchronous model, we model cell division as a random process occurring with the reaction rates

$$r_{g \to gg} = g \times g_{gg} \times \left( F_{\text{comp}} \times 1 \right)^{-1},$$

$$r_{g \to gs} = g \times g_{gs} \times \left( F_{\text{comp}} \times \left( 1 + c_{g \to s} \right) \right)^{-1},$$

$$r_{g \to ss} = g \times g_{ss} \times \left( F_{\text{comp}} \times \left( 1 + 2c_{g \to s} \right) \right)^{-1},$$

$$r_{s \to gg} = s \times s_{gg} \times \left( F_{\text{comp}} \times \left( 1 + 2c_{s \to g} \right) \right)^{-1},$$

$$r_{s \to gs} = s \times s_{gs} \times \left( F_{\text{comp}} \times \left( 1 + c_{s \to g} \right) \right)^{-1},$$

$$r_{s \to ss} = s \times s_{ss} \times \left( F_{\text{comp}} \times 1 \right)^{-1}, \tag{14}$$

where $s, g$ are the number of germ and soma cells in the organism, $s_{xy}, g_{xy}$ are elements of the differentiation program $D$, $F_{\text{comp}}$ is the composition effect profile computed identically to the synchronous model, see *Equation 3*, and $c_{s \to g}$ and $c_{g \to s}$ are differentiation costs.

We use the Gillespie algorithm to find which kind of cell division occurs next and how much time does it take. Then a chosen cell division occurs once (organism grows by a single cell). After that $F_{\text{comp}}$ value is updated to reflect the changed composition. Then the next cell division is sampled and the process continues until the organism reaches the maturity size. This model is designed to be the asynchronous implementation of our ideas, which remains close to our original model presented in the main text. Therefore, the rest of simulation protocol remains the same.

Computation time of the asynchronous model scales linearly with the number of cell divisions: it takes 1023 simulation steps to simulate the growth from 1 to 1024 cells. Therefore, it is computationally much more demanding than the synchronous model. The synchronous model scales linearly with the number of cell generations: the same growth to 1024 cells needs only 10 steps there. Maps similar to *Figure 2A–C* are unavailable with asynchronous model for computational reasons. Still, we calculated optimal differentiation strategies for a single combination of costs: $c_{g \to s} = 0$, $c_{s \to g} = 10$. Under these conditions, which favour evolution of irreversible differentiation in the synchronous model, it is significantly suppressed in the asynchronous model, see Fig. G.

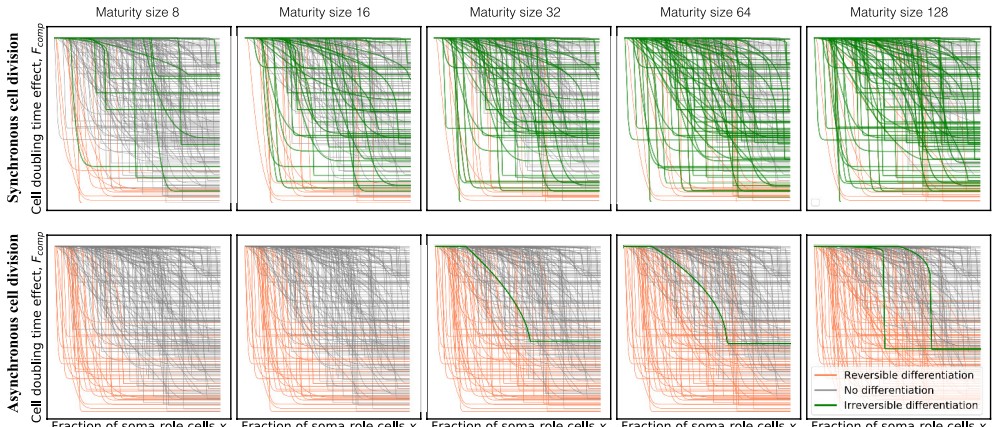

**Appendix 7—figure 1.** The model with asynchronous cell division suppresses evolution of irreversible somatic differentiation even at the most favourable conditions ($c_{g \to s} = 0$, $c_{s \to g} = 10$). Panels show composition profiles $F_{\mathrm{comp}}$ coloured according to the class of differentiation strategy: irreversible (green), reversible (orange), and no differentiation (black). In the synchronous model (top row), due to very asymmetric costs, irreversible differentiation is capable of evolving at small maturity sizes (eight cells). In the asynchronous model (bottom row), irreversible differentiation is not observed below 32 cells. Even at large sizes, the fraction of profiles promoting irreversible differentiation remains small.

The reason behind the difference between results for synchronous and asynchronous models is the different performance of reversible strategies in these models. If costs of soma differentiation are large enough, the expected period of cell division for a differentiating soma-role cell is longer than the length of life cycle. As a result, instead of re-differentiation, soma-role cells become effectively terminally differentiated. In such a situation, the growth of the organism is determined by propagation of germ-role cells and does not depend on the value of soma differentiation costs.

The key to success of reversible strategies in the synchronous model was an ability to develop large fractions of soma-role cells early on and to keep this fraction in the course of life cycle. There, the fraction of soma-role cells is preserved by a dynamic equilibrium between differentiation in both directions, see Appendix 2. In the asynchronous model with high soma differentiation costs, soma-role cells do not divide and such a dynamic equilibrium does not exist. The fraction of soma-role cells is maintained differently here. If we denote the number of germ-role and soma-role cells at time $j$ (unlike Appendix 2, it is a continuous parameter here) as $g(j)$ and $s(j)$, respectively, then in the case of non-dividing soma-role cells ($c_{g \to s} = 0$, $c_{s \to g} \gg 1$, $s_{ss} = 0$), the dynamics of the organism is given by

$$\frac{d}{dj}g(j) = 2(1 - m_g)g(j),$$

$$\frac{d}{dj}s(j) = 2m_g g(j), \tag{15}$$

The solution of this system of equations with initial condition of one germ-role and no soma-role cells is

$$g(j) = e^{2(1-m_g)j},$$

$$s(j) = \frac{m_g}{1 - m_g}\left(e^{2(1-m_g)j} - 1\right) \tag{16}$$

Hence, the fraction $r_s(j)$ of soma-role cells is

$$r_s(j) = \frac{m_g(1 - e^{-2(1-m_g)j})}{1 - m_g e^{-2(1-m_g)j}} \xrightarrow{j \to \infty} m_g. \tag{17}$$

The differentiation strategy considered above ($s_{ss} = 0$) is an extreme case where a dynamic equilibrium between cell differentiations is not possible. Still, *Equation 17* demonstrates that a balance between germ-role and soma-role cells is still achieved here. Therefore, in the asynchronous model with highly asymmetric differentiation costs, the reversible strategies keep all components that make them successful in the no costs scenario: the early production of soma-role cells due to high differentiation rates, the necessary fraction of soma-role cells during the life cycle (*Equation 17*), and the overall fast growth of the whole organism, despite having non-dividing soma-role cells (*Equation 16*).

Note that in irreversible strategies, soma-role cells do not differentiate and therefore divide at a normal rate. Therefore, the characteristic trade-off of irreversible strategies between having more soma-role cells early and more germ-role cells later in life cycle remains in place even in the asynchronous model. As a result, in this model, reversible strategies are not punished by asymmetric costs and outcompete irreversible ones.

