## [Decision Letter]

**Acceptance summary:**

The paper proposes a model, which studies an often-neglected aspect of cellular differentiation and division of labour. While the model is relatively simple, the premise and the findings are thought-provoking and this study can potentially provide the groundwork for further investigation.

**Decision letter after peer review:**

Thank you for submitting your article "Evolution of irreversible somatic differentiation" for consideration by *eLife*. Your article has been reviewed by 3 peer reviewers, and the evaluation has been overseen by a Reviewing Editor and Aleksandra Walczak as the Senior Editor. The following individuals involved in review of your submission have agreed to reveal their identity: E. Yagmur Erten (Reviewer #1); Guy Cooper (Reviewer #2); George Constable (Reviewer #3).

Essential revisions:

All reviewers found value in your work, stressed the simplicity and elegance of your model, and appreciated the insights and intuitions it provides. This elegance, however, comes at a cost. Your model relies on a key assumption, namely that cell divisions within an individual are synchronous, and that there does not seem to be within-host fitness differences between the different cell types. Reviewers questioned the biological relevance of the assumption, and we think it would be profitable to investigate further its impact on the model's results.

The discussion among reviewers also highlighted that the presentation of the model lacks details, and would need to be more precise and formalized. Reviewer #3, in particular, provides specific suggestions for the formalization of the model and leads for its analysis. I would like to encourage you to try and analyse the model, and at least to provide in the paper all elements necessary to fully understand it (in the form of equations, but also links to code). Notation should also be clarified, especially for the difference between time and cell generations, and add information about time in the notation (especially in the definition of c). The results are appealing and make intuitive sense, but careful readers need to be able to fully understand the model, and this is not the case with the current presentation of the manuscript. As the model is clarified, new questions may arise, which is why I am not making a recommendation for acceptance at this stage.

I encourage the authors to address all the comments made by the reviewers.

*Reviewer #1 (Recommendations for the authors):*

1. Lines 64-67: In what way the current study (model, assumptions etc.) differs from Cooper and West (2018), such that irreversible somatic differentiation is observed in this study but not in Cooper and West (2018)?

2. Lines 80-81: It is unclear at this point through what mechanism somatic cells accelerate growth. Do the organisms grow faster because somatic cells themselves divide at a faster rate, so having more of them means shorter development time? Or do the somatic cells contribute to overall resources available to all cells and every cell (including germ-role ones) divides faster? It becomes clearer later on and I think in their particular model it would not make a difference. But it would help to at least indicate that more explanation will come later.

3. Lines 125-128: The authors use a functional form (Equation 2) to determine soma cells' contribution to the growth rate. As their results depend on the shape of this function, I am wondering if there are empirical studies that support one type of form or the other. For instance, under what conditions would soma cells work better alone (Line 128)? In other words, which of these functional forms we are more likely to encounter in nature? This is later discussed to some extent, but references to the relevant literature (e.g. other models) could be useful in the Methods section as well, if a reader wanted to check other related approaches.

4. The authors refer to Appendix 3 for the first time at line 177, whereas while reading the results up to this point, I kept wondering what the fractions of the other strategies (RSD and NSD) were. In case adding the figures for RSD and NSD to the main text distracts from the main message, I think at least mentioning that they are at Appendix 3 much earlier in the Results section would help the readers.

5. Line 565: Here the authors say that large b favours ISD and a very large one promotes RSD, whereas in the main text they say "neither extremely large, nor extremely small" b favours ISD (Lines 208-209), which I found somewhat inconsistent.

6. It is not clear to me why the evolution of irreversible somatic differentiation requires a large enough organismal size. Also, in the main text, the authors do not mention what instead evolves in smaller organisms (RSD or NSD? This is later found in Appendix 3, but is not referred to or discussed in the main text). The authors later link their results about body size to some empirical examples in the Discussion section, but again, they do not discuss what might underlie these empirical observations or their findings about body size.

7. The second paragraph of the Discussion seems out-of-place as it is. I also cannot follow the logic; why do these cell numbers indicate organismal synchronicity? And what about cell death?

*Reviewer #2 (Recommendations for the authors):*

I like the model, it is simple and easy to interpret, providing predictions that make sense. However, it is not as general a model as the discussion implies in some cases. The predictions of the model are likely to depend on modelling assumptions that may be unrealistic in different systems, including the examples often cited in the paper.

My biggest request is that I would like more of a discussion of the limits that arise due to the these assumptions. In particular, to what extent are the predictions contingent on the fact that soma provide benefits continuously as the group grows? This is not the case for many of the systems cited in the work, such as in the Volvocine algae and in fruiting body formations such as in *Dictyostelium*. Furthermore, one could also imagine that differentiation probabilities are density dependent, or that germ cell fecundity depends on the number of soma cells in the last generation. I suspect that predictions 2 and 3 would not necessarily hold in these scenarios, which could explain for instance why many Volvocine species have a very large number of somatic cells. Acknowledging and discussing exactly how the predictions hinge on these assumptions would make the analysis much stronger.

Secondly, I think some definitions could be clearer in the introduction. For instance, if soma do not replicate at all, does it even make sense to speak of irreversible soma vs reversible soma? Many of the models cited have sterile soma that do not replicate (most Michod models, and Cooper and West model at least). Similarly, what if separation between germ and soma only occurs in one-generation of the group life cycle? What does the distinction between irreversible vs reversible soma mean in this case? Is irreversible soma just the same as soma sterility? How does all of this compare to the germline sequestration question, which readers may be more familiar with? These distinctions could be much clearer, which would help to set up the key question of the paper and make its scope more obvious.

Finally, I think some aspects of the presentation of the results could be improved. I found Figure 2A in isolation difficult to fully interpret. There are three outcomes in this model, ISD, RSD, and NSD, and the frequency of each outcome is only shown in Appendix 3. I would suggest including the frequency of the two other strategies in the main text. The same applies to Figure 4. You can't infer from just looking at the frequency of ISD alone to what extent the patterns are driven by irreversible soma being favoured over reversible soma vs no soma being favoured at all.

*Reviewer #3 (Recommendations for the authors):*

I very much enjoyed this paper, and only have a few suggestions with respect to the model.

I think potential conceptual limitations of the model lie in the assumptions of synchronous cell division and constant development strategy.

It may be possible to address the first of these issues (and thus the initial concern of Dr. Walczak) with some illustrative supplementary simulations (e.g. preliminary results to demonstrate the extent to which maturation time is affected by such asynchronicity). These might even take the form of some simple continuous time ODE models.

However the second of these issues would be a highly difficult task, and lies well outside the scope of the current paper. While exploring this question might certainly serve as a nice extension to the current work, I would not expect the authors to tackle this in the current context, where it would merely muddle the story presented.

Finally, while I like the model in general, there are some points of clarification I think could be made. Although I feel I have understood the core elements, there are some points of ambiguity where it is possible that I may be mistaken, and ironing out these potential misconceptions in the appendices would be beneficial for readers.

As I understand it:

The fraction of soma and germ cells in an organism are given by g(t) and s(t) in Equation 7, with s(t)=x in the main text (see Equation 2 – this should be made consistent?).

Note that 't' here refers to the generation t=1,2,…,n

These dynamics are independent of the costs of differentiation, c_g_ and c_s_.

However, the division time for cells in the organism during growth is dependent on these costs (see 't' in Equation 1 – note that 't' here is the continuous doubling time, which has an inconsistent notation with Equations 4-8).

Writing T_gen_(t) for the doubling time at generation t, we have

T_gen_(t)=F_diff_ * F_comp_

= (1+<c> )*( 1 – b + b ( ( x_1_ – s(t) )/( x_1_ – x_0_ ) ))^\α^

(when x_0_<s(t)<x_1_ – for simplicity I won't write out the other conditions)

At this point I'm not 100% sure how <c> is defined. I'd assume the following:

<c> = 1+s(t)*c_s_*(s_gs_ + 2 s_gg_)+g(t)*c_g_*(g_gs_ + 2 g_ss_)

(Is this correct?)

At this point I have T_gen_(t) as a function of t (having substituted for g(t) and s(t) from Equation 7). This allows me to write the maturation time (time to reach a size 2^n^) as

T_mat_ = \sum_t=1_^n^ T_gen_(t)

Finally, the fitness of an organism with a particular developmental strategy is given by the rate of gamete production (i.e. the number of gametes at maturity divided by the time taken to reach maturity)

W = g(n) / T_mat_

Working out the evolutionary optimal strategy is then a matter of maximising W with respect to s_gs_, s_ss_, g_gs_ and g_ss_.

Is this all correct?

If so, it may be possible to make analytical progress on this problem by replacing the discontinuous function in Equation 1 with a continuous approximation, e.g.

1 – (1 – b) x\^β^ / ( x\^β^ + (1 – x)\^β^ )

I mainly mention this latter point as a potential area of future investigation. However with respect to the model details, I would recommend the authors clarify the points above in one of the appendices.

[Editors' note: further revisions were suggested prior to acceptance, as described below.]

Thank you for submitting your article "Evolution of irreversible somatic differentiation" for consideration by *eLife*. Your article has been reviewed by 3 peer reviewers, and the evaluation has been overseen by Aleksandra Walczak as the Senior and Reviewing Editor. The following individuals involved in review of your submission have agreed to reveal their identity: E. Yagmur Erten (Reviewer #1); Guy Cooper (Reviewer #2); George Constable (Reviewer #3).

Essential Revisions:

All reviewers note that the paper has greatly improved. However they still would like to see some small changes made to improve clarity (no new research is required). Here is a summary but please go through the reviews attached below for concrete points:

– Clarify the details of the new models, as suggested by reviewer 1;

– Define soma and germ cells clearly, as noted by reviewer 2;

– Provide more discussion on the assumption of fixed strategies, as noted by reviewer 2;

– Add clarifications as suggested by reviewer 3.

*Reviewer #1 (Recommendations for the authors):*

I thank the authors for their responses and clarifications, as well as for extending their model to include risky differentiation and asynchronous cell division. I very much enjoyed reading the revised version of their manuscript, but I have the following questions about the new models they included.

Risky cell differentiation (Appendix 6): I don't exactly follow why the authors multiply the frequency and not the number of cell differentiations with per differentiation death risk. Maybe I am misunderstanding something, but isn't this implicitly assuming that deadly differentiation errors, if they happen in larger organisms, will have less probability to have an impact than in smaller organisms? Or in other words, the effect of deadly differentiation will be larger in smaller organisms? If true, this assumption might still be realistic, e.g. one deadly cell within 32 cells compared to 1024 can plausibly have a larger organismal level effect. But one can also argue that if one cell becomes a mutant and acquires a growth advantage, the overall size of the organism might not matter, especially e.g. if the mutant occurs early in the development or has a very high cell proliferation/resource uptake rate. Although this might not change the results in the main text qualitatively, as there the authors use one maturity size (2^10^) in their calculations.

Cell division asynchrony (Appendix 7): seeing the results of the cell division asynchrony, it seems like synchrony is almost a necessary condition for the evolution of irreversible differentiation in their model, just like those conditions summarized in Lines 335-339 albeit perhaps not as strict as them (since, although rarely, irreversible strategies still evolve). Perhaps it should be acknowledged as such earlier and more explicitly, rather than at the end of Discussion? Particularly given the fact that the authors looked at one of the most favourable conditions for irreversible strategies (c _(s-> g)_ >> c_(g->s)_) and found that the evolution of irreversible differentiation is very rare.

*Reviewer #2 (Recommendations for the authors):*

I think this model and analyses are very good and so I won't comment too much on these (with the exception of a thought on the asynchrony model). I think a few things need to be clarified but I am otherwise happy to endorse the paper for publication.

My main comments will concern the introduction and framing of this study as I think that there are still things that need to be made clear, which will really help the reader right from the start.

I think some very clear definitions of different terms are needed and as early as possible in the introduction. From what I can gather across different sections of the introduction: the soma cells are those that contribute to vegetative functions (sustaining the overall organism) but cannot act as a seed/propagule/spore for founding of a new organism. In contrast, germ cells are those that do not contribute to vegetative functions but can act as such a spore. The authors distinguish between terminally differentiated soma that do not divide such as in cyanobacteria and non-terminally differentiated soma that can divide such as in the Volvocales. They then ask the question in what conditions do the latter kind of soma (non-terminally differentiated) become irreversibly differentiated (that is when they can only divide and produce more soma). If the above is correct, I would suggest making these definitions and distinctions clearer and more localised rather than having bits of each definition spread across the introduction in a way that needs piecing together (perhaps a glossary type table could also help?)

If the above is correct, then the definitions need to be applied more consistently throughout the manuscript. For instance, the "somatic" cells that exist during the growth of the "higher" Volvocales do not qualify as somatic cells per the author's definition as they do not contribute to vegetative functions. In this case only the last generation of flagella beaters are "soma", none of which divide and so the distinction between reversibly and irreversibly differentiated does not apply here. The authors have added a paragraph about this in the discussion but they lean on the Volvocales so much in the introduction and discussion of their work that this mismatch needs to be flagged much sooner in the paper.

A similar issue applies to the discussion of Cooper and West 2018. Much as I would like to pretend that this paper could potentially cover all possibilities, group growth is not explicitly modelled here and so the distinction between reversible and irreversibly differentiated soma does not apply here (one can imagine that in this model there are no cooperative interactions as the group grows and that division of labour then may occur but only in the last generation of the group life cycle before spore dispersal much like for the Volvocales). If non-sterile helpers count as soma then this might be a different issue as sterile cells may be considered irreversible soma and non-sterile helpers as reversible soma, but then these non-sterile helpers can "seed" the next generation so I don't think they really qualify as soma per the author's definition? Having clearer definitions will help resolve these confusions.

Otherwise, I feel that the authors have sidestepped the potential impact of non-static traits in their model by saying in their response to reviewers that they have plans for a future paper on this. That is great and I am very much looking forward to what they find but this issue still needs to be discussed in this paper as many of their results here could be explained as arising directly from the assumption of static strategies as the group grows. I would suggest mentioning this at least once or twice as they go through the results (around lines 238-242 would be good) and then a whole paragraph on this in the discussion is warranted (can also mention plans for future work on this here). For instance, the need for just a few somatic cells that provide large benefits seems to arise directly from the fact that germ cells can't modulate the number of soma cells they spawn once these become too numerous, or that they can't have a time based strategy that produces many soma earlier but fewer later as the group grows.

I think the results they have found in the asynchronous model is really good but needs more explanation/discussion of why ISD can't seem to work here. They have modelled a time to replication cost as arising from the different differentiation costs. I find it strange in that case that RSD is not the worst affected strategy as the authors have established that this is the strategy with the most differentiation. I otherwise would have thought that having soma cells that divert their energy to vegetative functions as the slower replicator might have been a natural way to introduce asynchrony.

Finally, I think a word of caution on the discussion of "convex" shapes and how this favours division of labour/terminal differentiation/irreversible differentiation (lines 323-332). In several of the models cited (if not all), the convexity at issue is the relationship between an individual's investment in a public good/vegetative function and the fitness return to the group. In the authors' paper, the convexity is between the number/proportion of "helpers/soma" in the group and the fitness return to the group. These are very different things (one has to do with synergy from internal efficiencies whereas as the other comes from synergies from between individual interactions) and so should not be treated as the same prediction.

*Reviewer #3 (Recommendations for the authors):*

The authors have made substantial changes to the manuscript that appear to have addressed many of the concerns of the reviewers.

In their response, the reviewers clarified some details of the model, and I now feel I have a better understanding as to how it works. However that has led to another couple of small suggestions on my part that I believe would help readers.

In my original review I stated:

"Note that 't' here refers to the generation t=1,2,…,n

…

However, the division time for cells in the organism during growth is dependent on these costs (see 't' in Equation 1 – note that 't' here is the continuous doubling time, which has an inconsistent notation with Equations 4-8)"

I now see that under the costless differentiation assumed in Appendix 2 , t becomes an integer which helps simplify the subsequent analysis. It's worthwhile to make a note of this fact (before the sentence "Then, the expected fractions.…" would be an obvious potential place to mention this).

The authors response also makes clear at multiple points that cell divisions are stochastic:

"Since the differentiation program is stochastic, the costs of differentiation depend on the actual number of differentiation events happened in the course of growth, rather than probabilities like g_gs_.",

"Since the differentiation strategy is stochastic, the time to reach maturity (T_mat_) and the number of offspring at the last stage (g(n)) are random values, which we sample by repeatedly simulating the process of growth."

"However, since the outcomes of cell divisions are stochastic, the sampling of developmental trajectories has to reflect that and in our case it is done numerically."

I understand this. My comments, which I may not have articulated clearly in my initial review, were more aimed at asking how much understanding could be gained from alternatively taking a mean field approach. Indeed, this is precisely the approach the authors themselves take in Appendix 2, where they "consider the mathematical expectation of the composition". This leads to the obvious question – why can't a similar approach be used when differentiation is not costless?

Of course, I completely understand that stochasticity could be very important in a model such as this (where initial cell numbers are low), and it may be that such a mean-field approach leads to misleading results with respect to the prediction of the mean population growth rate. If this is the case, I think the authors should make a statement of this fact somewhere, perhaps with a reference to results in Gao et al., 2019 with respect to the differences between mean field and stochastic predictions.

Otherwise the authors have done a good job of clarifying my questions and addressing my concerns.

---

## [Author Response]

Essential revisions:All reviewers found value in your work, stressed the simplicity and elegance of your model, and appreciated the insights and intuitions it provides. This elegance, however, comes at a cost. Your model relies on a key assumption, namely that cell divisions within an individual are synchronous, and that there does not seem to be within-host fitness differences between the different cell types. Reviewers questioned the biological relevance of the assumption, and we think it would be profitable to investigate further its impact on the model's results.The discussion among reviewers also highlighted that the presentation of the model lacks details, and would need to be more precise and formalized. Reviewer #3, in particular, provides specific suggestions for the formalization of the model and leads for its analysis. I would like to encourage you to try and analyse the model, and at least to provide in the paper all elements necessary to fully understand it (in the form of equations, but also links to code). Notation should also be clarified, especially for the difference between time and cell generations, and add information about time in the notation (especially in the definition of c). The results are appealing and make intuitive sense, but careful readers need to be able to fully understand the model, and this is not the case with the current presentation of the manuscript. As the model is clarified, new questions may arise, which is why I am not making a recommendation for acceptance at this stage.I encourage the authors to address all the comments made by the reviewers.

Thank you and the reviewers for so well thought and constructive reviews. In the revised version of the manuscript, we additionally test our findings with a model featuring the risky differentiation (inspired by cancer) and an asynchronous cell division. We also clarified the model presentation. Later, next to the reviewers’ comments, we describe the specific changes in details.

Reviewer #1 (Recommendations for the authors):1. Lines 64-67: In what way the current study (model, assumptions etc.) differs from Cooper and West (2018), such that irreversible somatic differentiation is observed in this study but not in Cooper and West (2018)?

We discuss this in the updated manuscript. In principle, the ingredients we consider to be necessary for irreversible somatic differentiation are also included in that study, but the model setup is very different. For example, the sterile cells in Cooper and West do not divide further.

2. Lines 80-81: It is unclear at this point through what mechanism somatic cells accelerate growth. Do the organisms grow faster because somatic cells themselves divide at a faster rate, so having more of them means shorter development time? Or do the somatic cells contribute to overall resources available to all cells and every cell (including germ-role ones) divides faster? It becomes clearer later on and I think in their particular model it would not make a difference. But it would help to at least indicate that more explanation will come later.

Thank you for highlighting this issue and others below. We reworked the presentation of the model to make it clear. Regarding the specific question here, having soma-role cells allows for higher resource uptake by the organism, so having more soma-role cells generally results in faster developmental time. Since our main model is based on synchronous division, we do not allow that only somatic cells would divide faster.

3. Lines 125-128: The authors use a functional form (Equation 2) to determine soma cells' contribution to the growth rate. As their results depend on the shape of this function, I am wondering if there are empirical studies that support one type of form or the other. For instance, under what conditions would soma cells work better alone (Line 128)? In other words, which of these functional forms we are more likely to encounter in nature? This is later discussed to some extent, but references to the relevant literature (e.g. other models) could be useful in the Methods section as well, if a reader wanted to check other related approaches.

The choice of the functional form is motivated by its flexibility – with a right combination of parameters, a diverse scenarios can be covered: linear, concave, convex, a step threshold and so on. For instance, the prominent theoretical finding is that germ-soma differentiation arises when the tradeoff between viability and fertility is convex. Our functional form covers this set of models as a special case with x_0_ = 0 and x_1_=1. In the revised manuscript, we also discuss equivalent expressions used by other models.

Our statement of “soma cells working better alone” implied that there is no synergy between soma-role cells and increasing in the number of soma-role cells brings diminishing benefits. We now see that this wording causes confusion and rewrote that sentence.

4. The authors refer to Appendix 3 for the first time at line 177, whereas while reading the results up to this point, I kept wondering what the fractions of the other strategies (RSD and NSD) were. In case adding the figures for RSD and NSD to the main text distracts from the main message, I think at least mentioning that they are at Appendix 3 much earlier in the Results section would help the readers.

We also find figures for RSD and NSD very insightful but decided to put them into appendix to keep the focus of the paper on ISD. With great pleasure, we bring Figure 1 from appendix 3 to the main text.

5. Line 565: Here the authors say that large b favours ISD and a very large one promotes RSD, whereas in the main text they say "neither extremely large, nor extremely small" b favours ISD (Lines 208-209), which I found somewhat inconsistent.

Thank you for highlighting this. All fixed.

6. It is not clear to me why the evolution of irreversible somatic differentiation requires a large enough organismal size. Also, in the main text, the authors do not mention what instead evolves in smaller organisms (RSD or NSD? This is later found in Appendix 3, but is not referred to or discussed in the main text). The authors later link their results about body size to some empirical examples in the Discussion section, but again, they do not discuss what might underlie these empirical observations or their findings about body size.

Thank you for pointing on the lack of the discussion on this topic. In the revised manuscript, we discuss what causes the limitation of the organism size. In a nutshell, it is related to the monotonic increase in the fraction of soma cells in irreversible differentiation strategies and the decreased differentiation costs as the organism size increases.

7. The second paragraph of the Discussion seems out-of-place as it is. I also cannot follow the logic; why do these cell numbers indicate organismal synchronicity? And what about cell death?

There, we list two organisms, one with 32 and one with 128 cells. Both numbers are powers of two. If the cell divisions are synchronized at the level of organism, then at any moment, the number of cells should be power of two (cell death is not reported for these species). By contrast, in a bacterial colony founded by a single cell, cell divisions eventually desynchronize – at 32 cells this effect is already notable and by 128 cells it is apparent. Thus, we may suggest that some organism-level events synchronize cell division in the listed Volvocales. We polished this paragraph to be more clear and better integrated into the discussion.

Reviewer #2 (Recommendations for the authors):I like the model, it is simple and easy to interpret, providing predictions that make sense. However, it is not as general a model as the discussion implies in some cases. The predictions of the model are likely to depend on modelling assumptions that may be unrealistic in different systems, including the examples often cited in the paper.My biggest request is that I would like more of a discussion of the limits that arise due to the these assumptions. In particular, to what extent are the predictions contingent on the fact that soma provide benefits continuously as the group grows? This is not the case for many of the systems cited in the work, such as in the Volvocine algae and in fruiting body formations such as in *Dictyostelium*. Furthermore, one could also imagine that differentiation probabilities are density dependent, or that germ cell fecundity depends on the number of soma cells in the last generation. I suspect that predictions 2 and 3 would not necessarily hold in these scenarios, which could explain for instance why many Volvocine species have a very large number of somatic cells. Acknowledging and discussing exactly how the predictions hinge on these assumptions would make the analysis much stronger.

Thank you for raising this important topic. In the revised manuscript, we extend the discussion of our assumptions. We agree that our model is simplified comparing with the processes occurring in nature.

We assume that the soma-role cells provide benefits to the group growth during the whole ontogenesis. This may not accurately represent more complex species like *Volvox*, where daughter colonies grow within the maternal organism. Taking this into account can relax our last finding: “soma-role cells should bring benefits even in small numbers”, as only the final composition of the organism will matter in this case. However, a life cycle, where juvenile organisms are protected and nurtured by maternal body, represents quite an advanced degree of evolved complexity, which may not be directly relevant to the earliest stage of the evolution of multicellularity.

Secondly, I think some definitions could be clearer in the introduction. For instance, if soma do not replicate at all, does it even make sense to speak of irreversible soma vs reversible soma? Many of the models cited have sterile soma that do not replicate (most Michod models, and Cooper and West model at least). Similarly, what if separation between germ and soma only occurs in one-generation of the group life cycle? What does the distinction between irreversible vs reversible soma mean in this case? Is irreversible soma just the same as soma sterility? How does all of this compare to the germline sequestration question, which readers may be more familiar with? These distinctions could be much clearer, which would help to set up the key question of the paper and make its scope more obvious.

Thank you for highlighting this overlook. Indeed, the range of cell specializations that are called “soma cells” is very wide and we are looking in a particular type of these. By soma-role cells, we consider cells, which divide in the course of organism growth (unlike sterile/terminally differentiated cells), but do not contribute to the organism reproduction (unlike temporal division of labor).

We also do not consider the sequestration of the germ line and one-off separation between germ and soma, as these require considering a dynamical differentiation program, in which probabilities of differentiation change with time. We even considered such a model at the conceptualization stage but found an efficient but degenerate differentiation program – turn all cells into soma-role at the beginning of the life cycle and keep them until the end, where all cells turn to germ-role. This way, the developmental speed is extremely fast and the number of produced offspring is high as well. Obviously, this means that the space of dynamic differentiation programs is constrained but investigation of this space and its constraints is too far from the scope of the current work. We are planning to investigate this topic in later work.

Finally, I think some aspects of the presentation of the results could be improved. I found Figure 2A in isolation difficult to fully interpret. There are three outcomes in this model, ISD, RSD, and NSD, and the frequency of each outcome is only shown in Appendix 3. I would suggest including the frequency of the two other strategies in the main text. The same applies to Figure 4. You can't infer from just looking at the frequency of ISD alone to what extent the patterns are driven by irreversible soma being favoured over reversible soma vs no soma being favoured at all.

Thank you for the suggestion. Figure 1 from appendix 3 is elevated to the main text.

Reviewer #3 (Recommendations for the authors):I very much enjoyed this paper, and only have a few suggestions with respect to the model.I think potential conceptual limitations of the model lie in the assumptions of synchronous cell division and constant development strategy.It may be possible to address the first of these issues (and thus the initial concern of Dr. Walczak) with some illustrative supplementary simulations (e.g. preliminary results to demonstrate the extent to which maturation time is affected by such asynchronicity). These might even take the form of some simple continuous time ODE models.However the second of these issues would be a highly difficult task, and lies well outside the scope of the current paper. While exploring this question might certainly serve as a nice extension to the current work, I would not expect the authors to tackle this in the current context, where it would merely muddle the story presented.Finally, while I like the model in general, there are some points of clarification I think could be made. Although I feel I have understood the core elements, there are some points of ambiguity where it is possible that I may be mistaken, and ironing out these potential misconceptions in the appendices would be beneficial for readers.

Thank you for this feedback. With the help from your presentation, we rewrote the model section in the updated version of the manuscript. Below, there is our clarification of the model steps.

As I understand it:The fraction of soma and germ cells in an organism are given by g(t) and s(t) in Equation 7, with s(t)=x in the main text (see Equation 2 – this should be made consistent?).Note that 't' here refers to the generation t=1,2,…,nThese dynamics are independent of the costs of differentiation, c_g_ and c_s_.However, the division time for cells in the organism during growth is dependent on these costs (see 't' in Equation 1 – note that 't' here is the continuous doubling time, which has an inconsistent notation with Equation 4-8).Writing T_gen_(t) for the doubling time at generation t, we haveT_gen_(t)=F_diff_ * F_comp_= (1+<c> )*( 1 – b + b ( ( x_1_ – s(t) )/( x_1_ – x_0_ ) ))^\α^(when x_0_<s(t)<x_1_ – for simplicity I won't write out the other conditions)

Up to this point, all is correct.

At this point I'm not 100% sure how <c> is defined. I'd assume the following: <c> = 1+s(t)*c_s_*(s_gs_ + 2 s_gg_)+g(t)*c_g_*(g_gs_ + 2 g_ss_)(Is this correct?)

Not exactly. Since the differentiation program is stochastic, the costs of differentiation depend on the actual number of differentiation events happened in the course of growth, rather than probabilities like g_gs_. In each simulation, at each cell-generation step, we sample how many differentiation events (i.e. offspring is of a different type than the parent) will occur from a multinomial distribution. Each event brings c_s_ and c_g_ costs. The factor <c> used to compute the generation time is the average of these costs: the ratio between cumulative cost and the total number of cells at the beginning of the simulation step.

This way, the costs and the organism composition will differ among different runs of the same simulation. Hence, for each combination of control parameters, we run 300 independent realizations of organism growth and compute the expected population growth from this whole dataset (see below).

At this point I have T_gen_(t) as a function of t (having substituted for g(t) and s(t) from Equation 7). This allows me to write the maturation time (time to reach a size 2^n^) asT_mat_ = \sum_t=1_^n^ T_gen_(t)

T_mat_ is a sum of individual cell doubling times T_gen_ (t) but it is a random value from rather complex distribution and not a function.

Finally, the fitness of an organism with a particular developmental strategy is given by the rate of gamete production (i.e. the number of gametes at maturity divided by the time taken to reach maturity)W = g(n) / T_mat_

Since the differentiation strategy is stochastic, the time to reach maturity (T_mat_) and the number of offspring at the last stage (g(n)) are random values, which we sample by repeatedly simulating the process of growth. In our previous work (Interacting cells driving the evolution of multicellular life cycles, PloS CB, 2019), we have shown that a population with stochastic development will grow exponentially with the rate (W) given by the solution of equation

\sum_i_ P_i_ G_i_ e^-W T^_i_ = 1

where the sum is over possible trajectories i, P_i_ is the probability of each trajectory, G_i_ is the number of offspring produced at the maturity, T_i_ is the time to maturity. In our work, we numerically solve this equation using a sampled distribution of trajectories. In deterministic scenarios, like the absence of differentiation, the solution of this equation yields

W = log(g(n)) / T_mat_,

which has a similar form as your expression.

Working out the evolutionary optimal strategy is then a matter of maximising W with respect to s_gs_, s_ss_, g_gs_ and g_ss_.Is this all correct?

This is correct.

If so, it may be possible to make analytical progress on this problem by replacing the discontinuous function in Equation 1 with a continuous approximation, e.g.1 – (1 – b) x\^β^ / ( x\^β^ + (1 – x)\^β^ )

Thank you for a suggestion. However, since the outcomes of cell divisions are stochastic, the sampling of developmental trajectories has to reflect that and in our case it is done numerically. Unfortunately, a replacement of F_comp_ function with a continuous approximation does not lead to deeper analytical insights.

I mainly mention this latter point as a potential area of future investigation. However with respect to the model details, I would recommend the authors clarify the points above in one of the appendices.

Thank you again for careful consideration of our model. Now, we see that our initial presentation was confusing. In the updated text, we extended the presentation of the model.

[Editors' note: further revisions were suggested prior to acceptance, as described below.]

Reviewer #1 (Recommendations for the authors):I thank the authors for their responses and clarifications, as well as for extending their model to include risky differentiation and asynchronous cell division. I very much enjoyed reading the revised version of their manuscript, but I have the following questions about the new models they included.Risky cell differentiation (Appendix 6): I don't exactly follow why the authors multiply the frequency and not the number of cell differentiations with per differentiation death risk. Maybe I am misunderstanding something, but isn't this implicitly assuming that deadly differentiation errors, if they happen in larger organisms, will have less probability to have an impact than in smaller organisms? Or in other words, the effect of deadly differentiation will be larger in smaller organisms? If true, this assumption might still be realistic, e.g. one deadly cell within 32 cells compared to 1024 can plausibly have a larger organismal level effect. But one can also argue that if one cell becomes a mutant and acquires a growth advantage, the overall size of the organism might not matter, especially e.g. if the mutant occurs early in the development or has a very high cell proliferation/resource uptake rate. Although this might not change the results in the main text qualitatively, as there the authors use one maturity size (2^10^) in their calculations.

We appreciate your attention to details of the model design. During the manuscript revision, we discussed a lot how to implement the cell differentiation risk. The assumption, which you suggested – each individual defective cell brings the same risk to the organism, has also been closely considered. However, taking into account the context of our study, we settled on a different design. The key factor in our decision is that the organisms in our model constantly grow in size. As a result, the moment of the emergence of a defective cell plays a large role in the defect’s impact. If a defective cell emerges at the last round of cell divisions, it cannot harm the organism much, because the life cycle is about to end. However, a defective cell emerged in the very first cell division will have a large impact on the organism performance during the life cycle. To take this effect into account and make later defects less dangerous, we scaled the number of differentiation events by the organism size, thus attributing the risk to the frequency of differentiated cells.

We agree that a growth advantage of defective (defecting) cells may also play a role here. However, such an advantage should have much smaller impact on ever-growing simple organisms than on large animals. For animals, the division rate of the typical cell is close to zero, hence even a humble growth advantage of a defective cell brings a lot of difference to the cell dynamics over the span of long life cycle. In our case, every cell in the organism actively proliferates, while the life cycle is very short. Thus, the growth advantage must be large in order to have an impact. Hence, emergence of defects leading to a change in relative cell growth must be less likely in our model than in complex animals. On top of that, the rapid propagation of malignant tumors in human organism is often the end-result of a silent process of accumulation of multiple mutations, which can take years to complete. For the organisms we study, the defective cells just don’t have enough time to turn into proper cancer cells before the life cycle ends. Hence, we are convinced that the growth advantage of defective cells can be ignored in the context of our study.

We acknowledge that this design choice is tailored to simple organisms and may not be optimal for complex animals. In the revised manuscript, we added an exposition of our reasoning to the risky model section in Appendix 6.

Cell division asynchrony (Appendix 7): seeing the results of the cell division asynchrony, it seems like synchrony is almost a necessary condition for the evolution of irreversible differentiation in their model, just like those conditions summarized in Lines 335-339 albeit perhaps not as strict as them (since, although rarely, irreversible strategies still evolve). Perhaps it should be acknowledged as such earlier and more explicitly, rather than at the end of Discussion? Particularly given the fact that the authors looked at one of the most favourable conditions for irreversible strategies (c _(s-> g)_ >> c_(g->s)_) and found that the evolution of irreversible differentiation is very rare.

We agree that there is a need to mention the assumption of synchrony as early as possible. We now explicitly state that the model and the results are based on synchronous cell division in the Model section. And the asynchronous cell division is explored additionally in the Appendix 7.

Reviewer #2 (Recommendations for the authors):I think this model and analyses are very good and so I won't comment too much on these (with the exception of a thought on the asynchrony model). I think a few things need to be clarified but I am otherwise happy to endorse the paper for publication.My main comments will concern the introduction and framing of this study as I think that there are still things that need to be made clear, which will really help the reader right from the start.I think some very clear definitions of different terms are needed and as early as possible in the introduction. From what I can gather across different sections of the introduction: the soma cells are those that contribute to vegetative functions (sustaining the overall organism) but cannot act as a seed/propagule/spore for founding of a new organism. In contrast, germ cells are those that do not contribute to vegetative functions but can act as such a spore. The authors distinguish between terminally differentiated soma that do not divide such as in cyanobacteria and non-terminally differentiated soma that can divide such as in the Volvocales. They then ask the question in what conditions do the latter kind of soma (non-terminally differentiated) become irreversibly differentiated (that is when they can only divide and produce more soma). If the above is correct, I would suggest making these definitions and distinctions clearer and more localised rather than having bits of each definition spread across the introduction in a way that needs piecing together (perhaps a glossary type table could also help?)

We adopted the suggestion to clarify the differences between terminal differentiation and irreversible differentiation. We also explained the irreversible differentiation that is of most interest for us further in the Model section.

If the above is correct, then the definitions need to be applied more consistently throughout the manuscript. For instance, the "somatic" cells that exist during the growth of the "higher" Volvocales do not qualify as somatic cells per the author's definition as they do not contribute to vegetative functions. In this case only the last generation of flagella beaters are "soma", none of which divide and so the distinction between reversibly and irreversibly differentiated does not apply here. The authors have added a paragraph about this in the discussion but they lean on the Volvocales so much in the introduction and discussion of their work that this mismatch needs to be flagged much sooner in the paper.

Thank you for your thorough treatment of definitions! We define the germ-role as a state of a cell in which it continues to live after the organism fragments at the end of life cycle, and the soma-role as a state of a cell in which it dies upon fragmentation. We additionally assume that soma-role cells are capable to provide a benefit to the organism. However, this benefit is separate from the definition of soma-role itself. For example, according to our model, if the fraction of soma-role cells is below the contribution threshold x_0_, there is no benefit to the organism. Still, these cells are considered as soma-role. Hence, in your example, we see no conceptual problem with progenitors of flagella beaters not contributing to the vegetative functions, as contribution of soma-role cells is still possible under the right conditions (after the last round of cell divisions here) but not mandatory. In the revised manuscript we make our definitions more clear.

Our attention to Volvocales in the introduction and discussion reflects the overwhelming bias in the empirical literature about evolution of germ/soma differentiation towards this group (and also cellular slime molds but their multicellularity is aggregative). Without this group, simply, there is a little to discuss. Nevertheless, our model is not about Volvocales, we aim to answer the question about emergence of irreversible somatic differentiation in a broad context without tailoring it to the features of a single group. As a result, the design of our model does not fully reflect the features of Volvocales life cycle. We overlooked that our introduction made an impression that the manuscript is about a single prominent group. In the updated version, we modified the exposition of our work. We also explicitly acknowledge the bias of empirical examples in our discussion.

A similar issue applies to the discussion of Cooper and West 2018. Much as I would like to pretend that this paper could potentially cover all possibilities, group growth is not explicitly modelled here and so the distinction between reversible and irreversibly differentiated soma does not apply here (one can imagine that in this model there are no cooperative interactions as the group grows and that division of labour then may occur but only in the last generation of the group life cycle before spore dispersal much like for the Volvocales). If non-sterile helpers count as soma then this might be a different issue as sterile cells may be considered irreversible soma and non-sterile helpers as reversible soma, but then these non-sterile helpers can "seed" the next generation so I don't think they really qualify as soma per the author's definition? Having clearer definitions will help resolve these confusions.

We appreciate your suggestions. We have modified the comparison to this paper in the Introduction section. Meanwhile, as you mentioned in the above comment, we have elaborated the definition of germ-role and soma-role in the model section and stressed the differences between your and our models in discussion.

Otherwise, I feel that the authors have sidestepped the potential impact of non-static traits in their model by saying in their response to reviewers that they have plans for a future paper on this. That is great and I am very much looking forward to what they find but this issue still needs to be discussed in this paper as many of their results here could be explained as arising directly from the assumption of static strategies as the group grows. I would suggest mentioning this at least once or twice as they go through the results (around lines 238-242 would be good) and then a whole paragraph on this in the discussion is warranted (can also mention plans for future work on this here). For instance, the need for just a few somatic cells that provide large benefits seems to arise directly from the fact that germ cells can't modulate the number of soma cells they spawn once these become too numerous, or that they can't have a time based strategy that produces many soma earlier but fewer later as the group grows.

As you suggested, in the updated Results section, we acknowledge that our findings are obtained under assumption of static strategies. We also added a new paragraph to discussion about non-static differentiation strategies: their existence in nature and a possible impact on the model results.

I think the results they have found in the asynchronous model is really good but needs more explanation/discussion of why ISD can't seem to work here. They have modelled a time to replication cost as arising from the different differentiation costs. I find it strange in that case that RSD is not the worst affected strategy as the authors have established that this is the strategy with the most differentiation. I otherwise would have thought that having soma cells that divert their energy to vegetative functions as the slower replicator might have been a natural way to introduce asynchrony.

Irreversible differentiation is suppressed in the asynchronous model because there, the reversible strategies are capable to develop large number of soma-role cells early on and keep large fraction of germ-role cells throughout the life cycle even at the high soma differentiation costs (cs->g=10). Moreover, we would like to highlight that unlike in the synchronous model, here, high costs of soma differentiation bring an advantage to reversible differentiation strategies. The key effect here is that if costs of soma differentiation is large enough, the expected period of cell division for a differentiating soma-role cell is longer than the length of life cycle. As a result, instead of re-differentiation, soma-role cells become effectively terminally differentiated. In such a situation, the growth of the organism is determined by propagation of germ-role cells and is not affected by soma differentiation costs. Hence, the development of organisms using reversible strategies remains fast and irreversible strategies are outcompeted. In the updated manuscript, we discuss this effect in the Appendix 7.

We appreciate the idea of soma-role cells being slower replicators. We believe it may lead to a more elegant model. However, in this appendix, we test the impact of the assumption of the synchronous cell division used in the baseline model. Hence, it is more appropriate to present the asynchronous implementation of the base model keeping the remaining design intact. This immediately dictates that the costs must be associated with differentiation events and not with the cell state itself. Otherwise, it would be difficult to compare results from two models. We emphasize this motivation in the updated Appendix 7.

Finally, I think a word of caution on the discussion of "convex" shapes and how this favours division of labour/terminal differentiation/irreversible differentiation (lines 323-332). In several of the models cited (if not all), the convexity at issue is the relationship between an individual's investment in a public good/vegetative function and the fitness return to the group. In the authors' paper, the convexity is between the number/proportion of "helpers/soma" in the group and the fitness return to the group. These are very different things (one has to do with synergy from internal efficiencies whereas as the other comes from synergies from between individual interactions) and so should not be treated as the same prediction.

We agree with you on the analyses of the model differences between ours and the previous works. In the revised manuscript, we have stressed the model differences in the curvature of the shapes.

Reviewer #3 (Recommendations for the authors):The authors have made substantial changes to the manuscript that appear to have addressed many of the concerns of the reviewers.In their response, the reviewers clarified some details of the model, and I now feel I have a better understanding as to how it works. However that has led to another couple of small suggestions on my part that I believe would help readers.In my original review I stated:"Note that 't' here refers to the generation t=1,2,…,n…However, the division time for cells in the organism during growth is dependent on these costs (see 't' in Equation 1 – note that 't' here is the continuous doubling time, which has an inconsistent notation with Equations 4-8)"I now see that under the costless differentiation assumed in Appendix 2 , t becomes an integer which helps simplify the subsequent analysis. It's worthwhile to make a note of this fact (before the sentence "Then, the expected fractions.…" would be an obvious potential place to mention this).

Thanks – this was obviously a flaw in our notation. We have taken a new letter “j” to represent the previous “t” in the revised manuscript in Appendix 2 to clarify the number of cell divisions. We have further explained the variable “n”, which describes the maximal number of divisions. Thus j=1,⋯,n.

The authors response also makes clear at multiple points that cell divisions are stochastic:"Since the differentiation program is stochastic, the costs of differentiation depend on the actual number of differentiation events happened in the course of growth, rather than probabilities like g_gs_.","Since the differentiation strategy is stochastic, the time to reach maturity (T_mat_) and the number of offspring at the last stage (g(n)) are random values, which we sample by repeatedly simulating the process of growth.""However, since the outcomes of cell divisions are stochastic, the sampling of developmental trajectories has to reflect that and in our case it is done numerically."I understand this. My comments, which I may not have articulated clearly in my initial review, were more aimed at asking how much understanding could be gained from alternatively taking a mean field approach. Indeed, this is precisely the approach the authors themselves take in Appendix 2, where they "consider the mathematical expectation of the composition". This leads to the obvious question – why can't a similar approach be used when differentiation is not costless?Of course, I completely understand that stochasticity could be very important in a model such as this (where initial cell numbers are low), and it may be that such a mean-field approach leads to misleading results with respect to the prediction of the mean population growth rate. If this is the case, I think the authors should make a statement of this fact somewhere, perhaps with a reference to results in Gao et al., 2019 with respect to the differences between mean field and stochastic predictions.

Thank you for the clarification. While we completely agree that a mean field model can potentially provide a lot of insights about our findings, we have to admit that it does not do so. The expressions in the Appendix 2 are short and it seems promising to develop such a model further. However, once we put differentiation costs into play, things quickly get complicated. To find the population growth rate λ, we also need to compute the length of life cycle, which is not always possible. For the special case of step-function F_comp_, (e.g. x_0_=x_1_), ignoring the discreteness of the organism composition, and relying on Mathematica software, we managed to find the population growth rate λ explicitly. Shortening the notation with shortcut variables, it is given by:λ = {bM2(n+log⁡[1−xf])aM(1−b)x0−xf+β(Mn−(1−b)log⁡[mg→smg→s−Mx0]), x0<xfM(n+log⁡[1−xf])βn−αxf,x0>xfwhereM=mg→s+ms→gxf=mg→smg→s+ms→g(1−e−(ms→g+mg→s)n)α=csms→g−cgmg→sβ=mg→s+ms→g+mg→sms→g(cg+cs)

The condition x_0_<x_f_ checks whether the fraction of soma-role cells ever reaches the threshold value x_0_ (x_f_ is the fraction of soma-role cells at the end of life cycle, c.f. Equation 9 in Appendix 2).

Finding the optimal developmental strategies means finding the maximum of the expression above. This can be only performed numerically (and we already have a more detailed numerical model). In the absence of insightful results from this analytical approach, we just do not see the reason to include this result to the paper.

In our previous paper (Gao 2019) we did not use a mean field approximation (replacement of stochastic trajectories with the most likely one). Instead, our analytical results were obtained with the weak selection approximation (all stochastic trajectories have approximately similar duration Ti). We do not use this approximation in our current work because all the interesting strategies are found under such conditions where trajectories with many soma-role cells complete life cycle much faster, and thus, times to complete different trajectories differ a lot.